# (Amplified) Banded Matrix Factorization: A unified approach to private training

**Christopher A. Choquette-Choo**
Google DeepMind
cchoquette@google.com

**Arun Ganesh**
Google Research
arunganesh@google.com

**Ryan McKenna**
Google Research
mckennar@google.com

**H. Brendan McMahan**
Google Research
mcmahan@google.com

**Keith Rush**
Google Research
krush@google.com

**Abhradeep Thakurta**
Google DeepMind
athakurta@google.com

**Zheng Xu**
Google Research
xuzheng@google.com

## Abstract

Matrix factorization (MF) mechanisms for differential privacy (DP) have substantially improved the state-of-the-art in privacy-utility-computation tradeoffs for ML applications in a variety of scenarios, but in both the centralized and federated settings there remain instances where either MF cannot be easily applied, or other algorithms provide better tradeoffs (typically, as $\epsilon$ becomes small). In this work, we show how MF can subsume prior state-of-the-art algorithms in both federated and centralized training settings, across all privacy budgets. The key technique throughout is the construction of MF mechanisms with banded matrices (lower-triangular matrices with at most $\hat{b}$ nonzero bands including the main diagonal). For cross-device federated learning (FL), this enables multiple-participations with a relaxed device participation schema compatible with practical FL infrastructure (as demonstrated by a production deployment). In the centralized setting, we prove that banded matrices enjoy the same privacy amplification results as the ubiquitous DP-SGD algorithm, but can provide strictly better performance in most scenarios—this lets us always at least match DP-SGD, and often outperform it.

## 1 Introduction

We consider machine learning (ML) with DP in the centralized (datacenter) setting and the cross-device FL setting, extending and improving matrix factorization (MF) mechanisms[1] to advance the state-of-the-art in both. Given bounded-sensitivity batch gradients $\mathbf{x}_i \in \mathbb{R}^d$ for $i \in [n]$ steps, the MF-DP-FTRL algorithm uses a noise generation matrix $\mathbf{C}^{-1} \in \mathbb{R}^{n \times n}$ to return DP gradient estimates $\hat{\mathbf{x}}_i = \mathbf{x}_i + [\mathbf{C}^{-1}\mathbf{z}]_{[i,:]}$ where $\mathbf{z} \in \mathbb{R}^{n \times d}$ has IID entries $\mathcal{N}(0, \sigma^2)$ for suitable $\sigma > 0$. The noise correlation induced by $\mathbf{C}^{-1}$ is key to the success of MF-DP-FTRL. Alg. 1 and Sec. 2 provide details and intuition, and Table 1 in App. A summarizes notation and symbols.

---

[1]We use matrix factorization for the algorithmic design of DP mechanisms, and then use these mechanisms in general-purpose gradient-based private optimization algorithms. This use of matrix factorization is unrelated to machine learning settings such as collaborative filtering or recommender systems where the ML problem itself involves matrix factorization [37, 41, 51].

37th Conference on Neural Information Processing Systems (NeurIPS 2023).

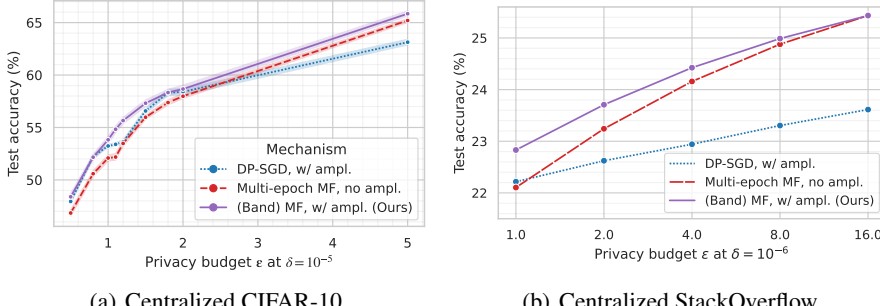

(a) Centralized CIFAR-10         (b) Centralized StackOverflow

Figure 1: **In the centralized setting, our BANDMF mechanism consistently performs at least as well as the best prior methods.** LEFT, (A): At $\epsilon \geq 0.5$, our BANDMF mechanism offers consistent utility benefits of around $1 - 4$ percentage points over either DP-SGD [1] or MULTI-EPOCH MF [15]. RIGHT, (B): BANDMF (bands $\hat{b} = 9, 18, 32,$ and $64$ for $\epsilon = 1-8$ respectively) significantly outperform both (unamplified) MULTI-EPOCH MF and amplified DP-SGD.

In **datacenter applications**, precise control of the sampling/shuffling of training data is possible, and so DP-SGD with privacy amplification [1] is one of the most popular ways to train machine learning models with formal privacy guarantees. However, Choquette-Choo et al. [15] recently demonstrated that a multi-epoch extension of the MF-DP-FTRL algorithm can outperform amplified DP-SGD in some settings, depending on the privacy and computational budget (typically larger budgets above $\epsilon \approx 2$ and a small number of training epochs). This leaves the state-of-the-art for centralized DP training in the unsatisfactory state where one must try both algorithms to be assured of the best performance.

In **cross-device federated learning**, devices choose when they are available to participate in training, and so precise sampling and shuffling is generally not possible (see Sec. 3 for more details). Motivated by these limitations which make amplified DP-SGD infeasible, Kairouz et al. [35] developed the (tree-aggregation-based) DP-FTRL algorithm. Their DP-FTRL does not rely on (or benefit from) privacy amplification and instead adds carefully correlated noise to the gradients to boost utility. Denisov et al. [17] proposed MF-DP-FTRL, replacing the tree-aggregation scheme of Kairouz et al. [35] with a general matrix-factorization mechanism. By optimizing over this space to find mechanisms with optimal error, substantial performance improvements were possible. However, the work of Denisov et al. [17] applies only to the single participation (single epoch) setting. Hence, for cross-device FL the state-of-the-art also requires considering multiple algorithms: tree-aggregation-based DP-FTRL when devices may participate more than one time, or MF-DP-FTRL when devices participate only once. Importantly, the extension of MF-DP-FTRL to multiple epochs of Choquette-Choo et al. [15] only applies in the centralized setting, as it again requires precise control of the participation pattern.

In this work, we address the limitations of MF-DP-FTRL (MF for short) noted above, and show that it can, in fact, achieve across-the-board state-of-the-art performance in both settings across all $\epsilon$. To accomplish this, we define a family of *banded* MF mechanisms, shown in Fig. 4 (c.f. Fig. 8 of App. D for visualizations of other factorization structures). We summarize our main contributions below.

**Contributions for cross-device FL**    Here, the $(k, b)$-participation schema of Choquette-Choo et al. [15] cannot be enforced. We propose a strict generalization, $b$-min-sep-participation, which can be practically enforced by FL infrastructure. We show how to efficiently and exactly bound the sensitivity for banded matrices in Thm. 2, allowing formal DP guarantees and the numerical optimization of optimal mechanisms (Sec. 4). These innovations lead to significant privacy-utility benefits in a production deployment (Fig. 6 of Sec. 6). Our work also generalizes the sensitivity calculations of Choquette-Choo et al. [15] to provide a general upper-bound on $b$-min-sep-participation sensitivity (Thm. 3), which allows the matrices of Choquette-Choo et al. [15] to be used in the FL setting, as well as removing the need to exactly bound $b$ before training (see Sec. 6 and App. K).

**Contributions for centralized training**    The existing privacy amplification analysis of DP-SGD does not allow for the correlated noise that is applied in MF-DP-FTRL. Our paper introduces a novel partitioning of the BANDMF iterates into independent queries. This allows us to prove in Thm. 4 of Sec. 5 that banded matrices enjoy the benefits of privacy amplification, and show that DP-SGD is a special case, giving us the best of both algorithms. This enables us to *always pareto-dominate DP-SGD*, unlike Choquette-Choo et al. [15] which only does so for large enough $\epsilon$ as observed in Fig. 1. Further, this allows us to improve on both baselines, between $1 - 4\%$-points. Informally:

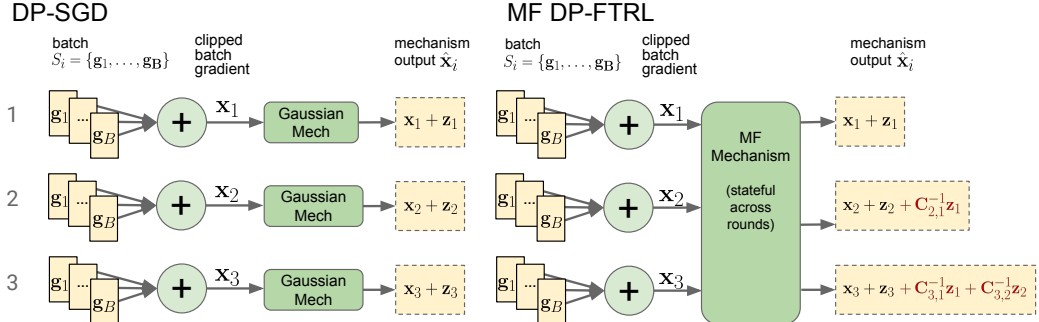

Figure 2: **MF-DP-FTRL (Alg. 1) enables noise cancelling across steps, where DP-SGD does not**. The entries $\mathbf{C}_{i,j}^{-1}$ are mostly negative (in $[0, -1)$) in matrices $\mathbf{C}^{-1}$ we consider (see Fig. 8). Thus, the red terms show that MF-DP-FTRL "cancels out" noise added on earlier iterations. For simplicity, we assume $\mathbf{C}^{-1}$ has 1s on the main diagonal and entries $\mathbf{C}_{i,j}^{-1}$ otherwise, with $\mathbf{z}_i := \mathbf{z}_{[i,:]}$ the rows of $\mathbf{z}$.

**Theorem 1** (Informal version of Theorems 4 and 5)**.** *Suppose we partition the dataset into $b$ equal-size subsets, and in step $i$ each example in the $i \pmod{b}$-th subset participates with probability $\frac{Bb}{m}$ where there are $m$ examples and the batch size is $B$. Then, a $\hat{b}$-banded $n$-iteration matrix mechanism with $\hat{b} \le b$ satisfies the same privacy guarantees as answering $n/b$ queries on a dataset, where each element of the dataset is independently included in each query with probability $\frac{Bb}{m}$.*

As an example of Thm. 1, consider doing $n = 2,000$ iterations of DP-SGD on CIFAR-10, which has $m = 50,000$ examples, using a minibatch of $B = 500$ examples in each round. This has the same DP guarantees as answering $2,000$ queries using a subsampled Gaussian mechanism with sampling probability $p = 500/50,000 = 0.01$. If we instead use, e.g., BANDMF with $\hat{b} = b = 10$, our suggested sampling scheme is the following: Partition CIFAR-10 into 10 subsets of $5,000$ examples each, $D_1, D_2, \ldots D_{10}$. In rounds 1, 11, 21... we sample 500 examples from $D_1$, in rounds 2, 12, 22... we sample 500 examples from $D_2$, and so on. We sample from each $D_i$ a total of $2000/10 = 200$ times, and each time our sampling probability is $500/5000 = 0.1$ within the subset. So Theorem 1 shows $\hat{b} = 10$ BANDMF satisfies the same DP guarantees as answering 200 queries with $p = 0.1$. As a special case of Theorem 1, DP-SGD is simply MF with a suitable diagonal matrix with $\hat{b} = 1$, and thus Thm. 1 recovers the privacy guarantees of DP-SGD with amplification by sampling. Empirically, we show that MF with amplification has privacy-utility tradeoffs that are *no worse than* DP-SGD *for all $\epsilon$, and often significantly better* as can be seen in Fig. 1.

Finally, we explore the computational tradeoffs of our approach. We find that banded matrices with $b$-min-sep-participation are equally efficient to optimize as those under $(k, b)$-participation but significantly reduce the memory and time complexity of the per-iteration noise generation from $\mathcal{O}(n)$ to a constant $\mathcal{O}(\hat{b})$ (where often total steps $n \gg d$). We will release all code with the final manuscript.

**Related work** The matrix mechanism (MF mechanism or MF) [40] has a rich history in offline, statistical queries [22, 28, 42, 62], with many applications including to online PCA [21], estimating marginals [22], and top-k selection [17]. Recently, this has been studied under the *adaptive streaming* setting, where privacy analysis must account for an adversary adaptively defining the inputs at each step [17, 24]. Denisov et al. [17] showed a connection with the DP-FTRL algorithm of Kairouz et al. [35] and with DP ML broadly; they showed that computing optimal MF significantly improves the privacy-utility-computation tradeoffs when making only a single pass (epoch) over the training data. Choquette-Choo et al. [15] showed that MF achieves state-of-the-art results in DP ML by showing how to optimize MF under arbitrary passes over the training data. Henzinger and Upadhyay [29] study the problem of DP-continual observation [13, 20] and the explicit factorization of the workload matrix $\mathbf{A}$ that minimizes the *completely bounded norm*, which is only off from the optimal by an additive constant. The connection between DP empirical risk minimization [3, 4, 5, 7, 8, 9, 14, 16, 23, 25, 32, 36, 39, 45, 52, 53, 54, 57] and DP online regret minimization [2, 4, 5, 26, 33, 35] has been studied for a long time. Asi et al. [5] demonstrated that DP-FTRL style algorithms [15, 17, 35] achieve the best known regret in certain classes of online learning problems (a.k.a. the realizable regime). An important question that still remains open is whether DP-FTRL style algorithms can obtain optimal population risk guarantees under DP [9].

**Algorithm 1** MF-DP-FTRL and DP-SGD

---

**Inputs:** Initial model $\boldsymbol{\theta}_1 \in \mathbb{R}^d$, dataset $D$ examples, matrix $\mathbf{C}^{-1} \in \mathbb{R}^{n \times n}$, noise $\mathbf{z} \in \mathbb{R}^{n \times d}$ with entries i.i.d. $\mathcal{N}(0, \sigma^2)$, clipnorm $\zeta$
  $\triangleright$ DP-SGD is simply the case $\mathbf{C}^{-1} = \mathbf{I}$, the $n \times n$ identity matrix.
**for** $i = 1, 2, \ldots, n$ **do**
    Select a set $S_i = \{\mathbf{d}_1, \ldots, \mathbf{d}_B\} \subseteq D$          $\triangleright$ Respecting schema $\Pi$, possibly sampling, Alg. 2
    $\mathbf{g}_j = \text{clip}(\nabla_\theta \text{loss}(\mathbf{d}_j, \boldsymbol{\theta}_i), \zeta)$ for $j \in [B]$, where $\text{clip}(\mathbf{d}, \zeta) = \min(1, \zeta/\|\mathbf{d}\|)\,\mathbf{d}$
    $\mathbf{x}_i = \sum_{j=1}^B \mathbf{g}_j$
    $\hat{\mathbf{x}}_i = \mathbf{x}_i + \zeta[\mathbf{C}^{-1}\mathbf{z}]_{[i,:]}$                        $\triangleright$ $\hat{\mathbf{x}}_i$ is now a DP estimate of $\mathbf{x}_i$
    $\boldsymbol{\theta}_{i+1} = \text{SGDM}(\boldsymbol{\theta}_i, \hat{\mathbf{x}}_i)$            $\triangleright$ Any first-order optimizer can be used in place of SGDM

---

## 2 Matrix Factorization, Sensitivity, and Efficient Implementations

Let $\mathbf{x} \in \mathbb{R}^{n \times d}$ be a stream of model gradients, and let $\mathbf{A} \in \mathbb{R}^{n \times n}$ be an appropriate linear query workload (prefix-sums, or a matrix encoding of stochastic gradient descent with momentum (SGDM) [17]). Matrix mechanisms use a factorization $\mathbf{A} = \mathbf{BC}$ to privately estimate the quantity $\mathbf{Ax}$ as

$$\widehat{\mathbf{Ax}} = \mathbf{B}\left(\mathbf{Cx} + \mathbf{z}\right), \tag{1}$$

where $\mathbf{z}$ is suitably calibrated noise to the sensitivity of the so-called 'query matrix' $\mathbf{C}$.

**Efficiently implementing MF-DP-FTRL**  Eq. (1) can be re-arranged as $\mathbf{A}(\mathbf{x} + \mathbf{C}^{-1}\mathbf{z})$. The multiplication by the linear operator $\mathbf{A}$ can now be viewed as post-processing of the noisy mechanism outputs $\mathbf{x} + \mathbf{C}^{-1}\mathbf{z}$; in many cases, this postprocessing has an efficient streaming implementation, e.g., simply passing gradients into a SGDM implementation. Thus, implementing MF-DP-FTRL is essentially equivalent to DP-SGD. The only difference is that the per-iteration gradient $\mathbf{x}_i$ is protected with noise $[\mathbf{C}^{-1}\mathbf{z}]_{[i,:]}$ rather than $\mathbf{z}_{[i,:]}$. Indeed, we see DP-SGD is a special case simply by taking $\mathbf{C} = \mathbf{C}^{-1} = \mathbf{I}$. Further, as long as the noise $\mathbf{C}^{-1}\mathbf{z}$ is computed correctly, the privacy guarantee holds, independent of the choice of $\mathbf{A}$. Alg. 1 gives the complete algorithm; with an appropriate choice of the matrix $\mathbf{C}^{-1}$, this algorithm captures DP-SGD, tree-aggregation DP-FTRL[2] [35], as well as MF-DP-FTRL [15, 17]. The multiplication of $\mathbf{C}^{-1}$ by Gaussian noise $\mathbf{z} \in \mathbb{R}^{n \times d}$ (which need never be fully materialized at once) is the critical step in the efficient implementation of MF. In App. J, we note this multiplication can be completed online for $\hat{b}$-banded matrices (defined formally in Sec. 3) in time and memory $\mathcal{O}(\hat{b}d)$ per training iteration compared to $\mathcal{O}(nd)$ for a non-banded matrix.

**Multiple participations**  We adopt the formalisms for multiple-participations of Choquette-Choo et al. [15]. We assume there are $m$ examples (or users in FL) in the database where $B$ examples are selected on each step $i \in [n]$. These $B$ chosen examples are said to *participate* on this step $i$. The examples at each step are processed via any adaptive function, e.g., computing a gradient of the current model (which depends on the model parameter values) as in Alg. 1. The per-example output vectors $\mathbf{g} \in \mathbb{R}^d$ are each bounded to $\ell_2$ norm at most $\zeta$ (noting that our notions of sensitivity scale linearly in $\zeta$, without loss of generality (WLOG) we take to be 1 in analysis below). These clipped vectors are then summed to yield $\mathbf{x}_i = \sum_{j=1}^B \mathbf{g}_j$. The MF-DP-FTRL mechanism releases the privatized estimates of $\mathbf{x}_i$ in a streaming fashion. The multi-epoch setting occurs when $m < n \cdot B$, so that every example necessarily participates more than once.

**Intuition for (anti-)correlated noise in MF-DP-FTRL**  Fig. 2 compares DP-SGD and MF-DP-FTRL. To gain an intuition for why MF-DP-FTRL can perform better than DP-SGD, observe that vanilla SGD has iterates $\theta_t = \theta_0 - \eta \sum_{i=1}^t \hat{\mathbf{x}}_i$, and hence when the noisy gradients $\hat{\mathbf{x}}_i$ are added, the $\mathbf{C}^{-1}_{[i,j]}\mathbf{z}_{[i,:]}$ terms in MF-DP-FTRL serve to *cancel out* some of the noise introduced on previous rounds. This reduces the total error in the final model (i.e., the prefix sums). However, this worsens the sensitivity of the mechanism to >1 (as it is for DP-SGD). This is because an adversary trying to learn $\mathbf{x}_1$ via $\hat{\mathbf{x}}_1$ can partially learn the value of $\mathbf{z}_1$ from $\hat{\mathbf{x}}_2$, whereas in DP-SGD $\hat{\mathbf{x}}_1$ and $\hat{\mathbf{x}}_2$ are uncorrelated. This tradeoff is what MF-DP-FTRL aims to minimize. More details on this intuition are in App. A.1.

---

[2]In this case we use a suitable pseudo-inverse $\mathbf{C}^\dagger \in \mathbb{R}^{n \times 2n-1}$ with noise $\mathbf{z} \in \mathbb{R}^{2n-1 \times n}$.

**Adjacency and participation schemas**   DP requires a notion of adjacent datasets. Two data streams $\mathbf{x}$ and $\tilde{\mathbf{x}}$ are adjacent if the data associated with any single example is altered, but not when this example participated. Thus, any $\mathbf{x}_i$ where example $\mathbf{d}_j$ participated can be changed subject to the constraint $\|\mathbf{g}_j^{(i)}\| \le \zeta$. However, the participation pattern does not change. A *participation schema* $\Pi$ gives the set of possible *participation patterns* $\pi \in \Pi$, with each $\pi \subseteq [n]$ indicating a set of steps in which a single example might participate. Let $\mathbf{N}$ be the set of all pairs of neighboring streams $\mathbf{x}$ and $\mathfrak{D} := \{\mathbf{x} - \tilde{\mathbf{x}} \mid (\mathbf{x}, \tilde{\mathbf{x}}) \in \mathbf{N}\}$ represent the set of all possible deltas between neighboring $\mathbf{x}, \tilde{\mathbf{x}}$. We say a $\mathfrak{D}$ **satisfies the participation schema** $\Pi$ if the indices of all nonzero rows in each $\mathbb{R}^{n \times d}$ matrix $\mathbf{u} \in \mathfrak{D}$ are a subset of some $\pi \in \Pi$. To illustrate this, single-participation is represented as $\Pi = \{\{1\}, \{2\}, \dots \{n\}\}$) and full-batch gradient descent (every-step) as $\Pi = \{[n]\}$). Choquette-Choo et al. [15] studied *fixed-epoch-order participation*, denoted $(k, b)$-*participation*, where each example participates at most $k$ times, with any adjacent participations exactly $b$ steps apart: formally, $\Pi$ is the set of all $\pi$ such that $|\pi| \le k$, and if $\pi = \{i_1, \dots, i_k\}$ indexed in increasing order, we have $\forall j \in \{2, \dots, k\}, i_j - i_{j-1} = b$. For example $(k=2, b=3)$-participation has $\Pi = \{\{1, 4\}, \{2, 5\}, \{3, 6\}\}$. As discussed in Choquette-Choo et al. [15], this setting faithfully captures centralized multi-epoch ML training setups with single and every-step as special cases. We can now define the **sensitivity** of the matrix factorization mechanism as

$$\mathrm{sens}_\Pi(\mathbf{C}) = \sup_{(\mathbf{x}, \tilde{\mathbf{x}}) \in \mathbf{N}} \|\mathbf{C}\mathbf{x} - \mathbf{C}\tilde{\mathbf{x}}\|_F = \sup_{\mathbf{u} \in \mathfrak{D}} \|\mathbf{C}\mathbf{u}\|_F. \tag{2}$$

**Optimizing factorizations**   Different factorizations $\mathbf{A} = \mathbf{B}\mathbf{C}$ can have very different performance in practice. Thus, in MF applications it is common to optimize over the space of factorizations, where the objective function is the expected total squared error on $\mathbf{A}$, given as $\mathcal{L}(\mathbf{B}, \mathbf{C}) = \mathrm{sens}_\Pi^2(\mathbf{C}) \|\mathbf{B}\|_F^2$. We define the expected root-mean-squared error (RMSE) as $\sigma\sqrt{\mathcal{L}(\mathbf{B}, \mathbf{C})/n}$, where $\sigma$ is the standard deviation of the Gaussian noise. We take $\sigma = 1$ when simply comparing mechanisms, or (in Sec. 6), calibrate $\sigma$ to achieve specific $(\epsilon, \delta)$-DP guarantees.

To facilitate optimization, utilizing the fact that the optimal-for-squared-error decoder $\mathbf{B}$ is $\mathbf{A}\mathbf{C}^\dagger$ [17], we note $\mathcal{L}(\mathbf{B}, \mathbf{C}) = \mathcal{L}(\mathbf{A}\mathbf{C}^\dagger, \mathbf{C})$. The expected total squared error is invariant to scaling $\mathbf{C}$ by a constant, and hence it is sufficient to optimize under a sensitivity 1 constraint. Further expressing the sensitivity and error in terms of $\mathbf{X} = \mathbf{C}^\top\mathbf{C}$ (note $\mathbf{X}$ is unrelated to the data $\mathbf{x}$), we have

$$\mathcal{L}(\mathbf{B}, \mathbf{C}) = \mathcal{L}(\mathbf{A}\mathbf{C}^\dagger, \mathbf{C}) = \|\mathbf{A}\mathbf{C}^\dagger\|_F^2 = \mathrm{tr}[\mathbf{A}^\top\mathbf{A}\mathbf{X}^{-1}], \tag{3}$$

assuming $\mathrm{sens}_\Pi(\mathbf{C}) = 1$ and $\mathbf{A}$ is in the rowspace of $\mathbf{C}$. Thus, we arrive at:

**Problem 1.** *The matrix factorization optimization problem is to solve the convex optimization*

$$\underset{\mathbf{X} \in \mathbf{S}_+^n}{minimize} \quad \mathrm{tr}[\mathbf{A}^\top\mathbf{A}\mathbf{X}^{-1}] \qquad subject\ to \quad \mathrm{sens}_\Pi^2(\mathbf{X}) \le 1, \tag{4}$$

*and then find $\mathbf{C}$ so that $\mathbf{C}^\top\mathbf{C} = \mathbf{X}$, e.g., via Cholesky decomposition.*

## 3   A Participation Schema for FL and the Sensitivity of Banded Matrices

In cross-device FL, devices locally evaluate eligibility criteria to deterimine when they might participate in training [10, 34], for example only checking-in to the coordinating server when they are plugged in, on unmetered wifi, and idle. This makes it practically difficult to enforce the $(k, b)$-participation of Choquette-Choo et al. [15], where devices are assumed to participate at the same relative position in each epoch: devices are unlikely to meet the eligibility criteria during the narrow windows of both step $i$ and $i + b$. Further, precise sampling cannot provide the same level of privacy amplification as in the centralized setting. Consider if 6500 devices are needed to complete a round [59]. An extreme (but realistic depending on time of day) setting may have only 6500 devices meeting eligibility criteria. Thus, either the protocol proceed without any sampling/amplification or wait until more devices are available; neither are desirable. We avoid amplification in the cross-device setting and instead proceed by addressing the question: *Can MF-DP-FTRL be extended to the cross-device federated learning setting with multiple client participations?*

With $(k, b)$-participation difficult to enforce in practice, our first challenge is to define a new participation schema with several properties: (a) the sensitivity of any matrix mechanism under this query can be bounded; (b) this bound is tight over an expressive class of matrices; (c) this bound

can be efficiently represented as a constraint in a mathematical program so as to be able to find a near-optimal factorization $\mathbf{A} = \mathbf{BC}$. In Defn. 1, we propose $b$-min-sep-participation, a generalization of $(k, b)$-participation which can be practically enforced by cross-device FL systems, thus enabling us to leverage BANDMF in this setting (see Sec. 6).[3] In $b$-min-sep-participation, the distance between any two participations is *at least* $b$, rather than *exactly* $b$ as in $(k, b)$-participation:

**Definition 1.** *The* $b$-***min-sep-participation*** *schema is given by*

$$\Pi_b = \left\{ \pi \subseteq [n] \mid \{i, j\} \subseteq \pi, i \neq j \Rightarrow |i - j| \geq b \right\}.$$

Observe this participation schema is easy for devices to enforce: each device remembers the last step $i$ in which it participated, and when it again becomes eligible, it checks in to the server, and participates in training as long as the current step is at least $i + b$; it does not need to check in during a narrow (and unknown to the device) time window for a specific step.

We now turn to computing sensitivity under $b$-min-sep-participation. For $(k, b)$-participation, $|\Pi_{(k,b)}| = b$, a fact Choquette-Choo et al. [15, Eq. 3] critically exploited when computing sensitivity via brute force computation of a maximum over the elements in $\Pi$. With $b$-min-sep-participation, we have $|\Pi_b| = \mathcal{O}(\exp(n))$, and hence any brute force approach which requires checking some value for all $\pi \in \Pi_b$ will be impractical. Following the formalism of [15, Section 2], a participation schema $\Pi$ (plus a specification of model dimension $d$) yields an expression for the sensitivity of the function $\mathbf{x} \mapsto \mathbf{Cx}$ assuming that the contributions of any given user to the data structure $\mathbf{x}$ are restricted to the rows in $\mathbf{x}$ indexed by some $\pi \in \Pi$. By Prop. E.1 of App. E.2, independent of model dimension $d$, we show sensitivity for *any* schema $\Pi$ may be bounded by

$$\mathrm{sens}_\Pi (\mathbf{C})^2 \leq \max_{\pi \in \Pi} \sum_{i,j \in \pi} |\mathbf{X}_{[i,j]}|. \tag{5}$$

Eq. (5) highlights several subtleties in computing sensitivity. First, is the challenge presented by the exponentially large number of patterns in $\Pi_b$. Second is the question of tightness of the inequality in Eq. (5): how much are we losing by effectively ignoring any cancellation in the matrix $\mathbf{X}$?

**Banded matrices** Fortunately, banded matrices render Eq. (5) both exactly computable and tight (independent of dimension $d$), showing that $\Pi_b$ satisfies the requirements of (b) and (c) above. We say a (general) matrix $\mathbf{X}$ is $\hat{b}$-banded if for all $i, j \in [n], |i - j| \geq \hat{b}$ implies $\mathbf{X}_{[i,j]} = 0$. While this is off-by-one from the bandwidth ($\mathbf{X}$ has bandwidth $b - 1$), our definition will be useful as it will be natural to match $\hat{b}$-banded matrices with $b$-min-separation. Further, for $\hat{b}$-banded lower-triangular matrices (which will play a central role), $\hat{b}$ intuitively gives the number of bands in the matrix.

For non-banded matrices, the right-hand side of Eq. (5) remains efficiently computable (but not easily expressible in a mathematical program), enabling us to provide nontrivial privacy guarantees for matrices which are not $b$ banded under $b$-min-sep, showing that $\Pi_b$ satisfies (a) as well. The key subroutine is Alg. 3, which gives an efficient dynamic program for solving linear optimization over $\Pi_b$. Define $\mathbf{u}(\pi) \in \{0, 1\}^n$ by $\mathbf{u}(\pi)_i = 1$ if $i \in \pi$ and 0 otherwise. Then, Alg. 3 solves

$$\min_{\pi \in \Pi_b} \langle \mathbf{v}, \mathbf{u}(\pi) \rangle.$$

This is the key subroutine in Alg. 4 and Alg. 5. Proofs for Thm. 2 and Thm. 3 are deferred to App. E.2.

**Theorem 2.** *Let* $\mathbf{C} \in \mathbb{R}^{n \times n}$ *be a lower-triangular matrix, and* $\Pi_b$ *the* $b$-min-sep-participation *schema. Further, suppose* $k'$ *upper-bounds the actual maximum number of participations that occurred in a data stream* $\mathbf{x}$ *(at worst, we may take* $k' \leq \lceil \frac{n}{b} \rceil$*): Then: (1) If* $\kappa$ *is an upper-bound on the column norms of* $\mathbf{C}$*, that is* $\forall j \in [n]$*,* $\|\mathbf{C}_{[:,j]}\| \leq \kappa$*, and* $\mathbf{C}$ *is* $b$-banded, then the sensitivity is bounded by $\kappa \sqrt{k'}$*. (2) If* $\mathbf{C}$ *is* $b$-banded, Alg. 5 invoked with Gram matrix $\mathbf{X} = \mathbf{C}^\top \mathbf{C}$ and $b, k'$ as in the setup, exactly computes* $\mathrm{sens}(\mathbf{C})$ *under schema* $\Pi_b$ *in polynomial time for any dimension* $d$*.*

**Theorem 3.** *For an arbitrary (non-banded)* $\mathbf{C}$*, let* $\mathbf{X} = \mathbf{C}^\top \mathbf{C}$ *and* $b, k'$ *as in Thm. 2. Then Alg. 4 of App. E upper-bounds* $\mathrm{sens}(\mathbf{C})$ *under schema* $\Pi_b$ *in polynomial time for any dimension* $d$*.*

---

[3]While $b$-min-sep-participation is required for the cross-device FL setting, $(k, b)$-participation is preferred in centralized settings where it can be enforced and will also satisfy our BANDMF amplification guarantees leading to the empirical results we show in Fig. 1.

## 4 Optimizing Banded Matrices

To enjoy the benefits of banded matrices within the framework of MF-DP-FTRL, we need to design an algorithm that can efficiently optimize over the space of $\hat{b}$-banded $\mathbf{C}$ matrices. To solve this problem, we will work in the domain of $\mathbf{X} = \mathbf{C}^\top \mathbf{C}$, and utilize the following fact:

**Proposition 4.1.** *Let $\mathbf{X} \in \mathbb{R}^{n \times n}$ be a $\hat{b}$-banded symmetric positive definite matrix. Then there exists a lower triangular $\hat{b}$-banded matrix $\mathbf{C} \in \mathbb{R}^{n \times n}$ such that $\mathbf{X} = \mathbf{C}^\top \mathbf{C}$.*

Utilizing Prop. 4.1, we can modify Problem 1 by introducing the constraint $\mathbf{X}_{[i,j]} = 0$ if $|i - j| \geq \hat{b}$. This additional linear constraint preserves convexity of the optimization problem, and makes the sensitivity calculation tractable as well. However, it is still not immediately obvious how to solve the optimization problem, since we need to run the dynamic program defined in Alg. 5 of App. E to compute sensitivity. For this reason, we impose the additional constraint that $\mathrm{diag}\,(\mathbf{X}) = \mathbf{1}$. This constraint, together with bandedness, ensures that the squared sensitivity is equal to $k$ for all $\mathbf{X}$ by Thm. 2. The final optimization problem we seek to solve is stated below:

**Problem 2.** *The matrix factorization optimization problem for banded matrices is to solve*

$$\underset{\mathbf{X} \in \mathbf{S}_+^n}{minimize}\ \mathrm{tr}[\mathbf{A}^\top \mathbf{A} \mathbf{X}^{-1}] \qquad subject\ to \quad \mathrm{diag}\,(\mathbf{X}) = \mathbf{1}\ and\ \mathbf{X}_{[i,j]} = 0\ if\ |i - j| \geq \hat{b}, \qquad (6)$$

*and then find $\mathbf{C}$ so that $\mathbf{C}^\top \mathbf{C} = \mathbf{X}$ via Prop. 4.1.*

We would like to remark on the similarity between Problem 2 and the single-participation version of Problem 1. The two problems are identical modulo the bandedness constraint, which is an equality constraint on individual entries of $\mathbf{X}$. Therefore, existing primal-optimization based solvers [43] for the single-participation matrix mechanism can be extended to optimize over this new space of matrices with little modification. Specifically, the only modification necessary is to initialize to an appropriately banded feasible $\mathbf{X}$ matrix, like $\mathbf{X} = \mathbf{I}$, and to post-process the gradient w.r.t $\mathbf{X}$ by setting $\frac{\partial L}{\partial \mathbf{X}_{[i,j]}} = 0$ if $|i - j| \geq \hat{b}$ in each step. Since the equality constraints exactly specify individual entries of $\mathbf{X}$, Problem 2 can be solved as an unconstrained optimization problem (over the remaining entries in $\mathbf{X}$), using any number of off-the-shelf unconstrained optimization algorithms.[4] As recommended by McKenna et al. [43], we use the LBFGS algorithm [12] to solve this problem.

**Remarks on the $\mathrm{diag}\,(\mathbf{X}) = \mathbf{1}$ constraint**   The constraint on $\mathrm{diag}\,(\mathbf{X})$ serves multiple purposes. First, $\mathrm{diag}\,(\mathbf{X}) = \mathbf{1}$ implies that $\|\mathbf{C}_{[:,i]}\|_2 = 1$ for all $i$, i.e., that $\mathbf{C}$ has equal column norms. This ensures that BANDMF reduces to DP-SGD when $\hat{b} = 1$, which is desirable. Second $\mathrm{diag}\,(\mathbf{X}) = \mathbf{1}$ simplifies the optimization problem greatly, as the sensitivity computation for both $(k, b)$-participation and $b$-min-sep are trivial and tight under this constraint (Thm. 2 Claim (1)). Third, imposing this constraint does not drastically change the search landscape, or cost much in terms of RMSE; see Table 2 for a comparison of matrices with and without this constraint, and Fig. 8 for a visualization. Fourth, this constraint allows us to solve a single optimization problem that is simultaneously tailored for $(k, b)$-participation and $b$-min-sep-participation. In Appendices B and C, we formulate an optimization problem without the $\mathrm{diag}\,(\mathbf{X}) = \mathbf{1}$ constraint, discuss how we solve it, and compare matrices generated with and without this constraint empirically.

## 5 Amplification for Banded Matrix Mechanisms

In the centralized setting where we can control the participation patterns of individual examples, the privacy guarantees of BANDMF can be amplified. We focus on amplification by sampling with fixed batch size in this section, but give a more general statement in App. F.

Existing privacy analysis of MF-DP-FTRL is based on the reduction to the batch release of the entire $\mathbf{C}\mathbf{x} + \mathbf{z}$ as a single Gaussian mechanism event [15, 17] so standard amplification techniques don't directly apply. Instead, for each participation by an example, we consider the set of rows in $\mathbf{C}\mathbf{x}$ affected by this participation as a Gaussian mechanism (see the groups of rows in Fig. 4). Then as

---

[4]Note that the constraint that $\mathbf{X}$ is positive definite is typically handled implicitly by taking sufficiently small step sizes to avoid violating that constraint [43, 62].

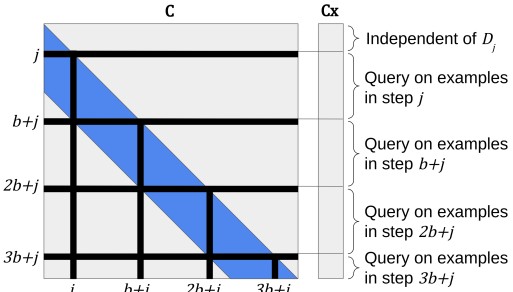

**Algorithm 2** Sampling scheme

---

$D_1, \ldots, D_b \leftarrow$ arbitrary partition of $D$.
**for** $i = 1, 2, \ldots, n$ **do**
    $j = i \pmod{b}$ ($b$ if $i/b$ is integer).
    $S_i \leftarrow$ random size $B$ subset of $D_j$.
    Compute $\mathbf{x}_i$ on $S_i$.
Release $\mathbf{C}\mathbf{x} + \mathbf{z}$.

---

Figure 3: An example of our sampling scheme that gives privacy amplification for BANDMF.

Figure 4: Visualization of how we can decompose BANDMF into independent queries when using Alg. 2. Larger view in Fig. 10 of App. F.

long as the sets of rows corresponding to different participations do not interact, which is ensured by the bandedness of $\mathbf{C}$, we can apply amplification to them separately.

Observe from Fig. 4 that the structure of BANDMF guarantees that the set of rows of $\mathbf{C}\mathbf{x}$ which depend on each of $\mathbf{x}_j, \mathbf{x}_{\hat{b}+j}, \mathbf{x}_{2\hat{b}+j}, \ldots$ are disjoint sets. Thus, we use the following sampling scheme for determining which examples participate in each step which is made formal as Alg. 2. Let $D_1, D_2, \ldots, D_{\hat{b}}$ be an arbitrary partition of $D$ into $\hat{b}$ indexed subsets of size $\check{m} \coloneqq \lfloor m/\hat{b} \rfloor$ (for simplicity, we discard extra examples so all $D_j$ have size exactly $\check{m}$). In steps $j, \hat{b} + j, 2\hat{b} + j, \ldots$, we will only use examples in $D_j$. Hence, participation follows $(k, b)$-participation for $b = \hat{b}$;[5] because it is optimal for $(k, b)$-participation to have the number of bands $\hat{b}$ equal the min-seperation $b$, in the remainder of this section and the associated appendices we simply write $b$ instead of $\hat{b}$. Within each of these steps, we sample a size $B$ subset of $D_j$ uniformly at random to use in computing $\mathbf{x}$.

Roughly speaking, Thm. 4 below shows that if we use Alg. 2, BANDMF satisfies any standard privacy guarantees satisfied by DP-SGD run for $k$ rounds, where in each round we sample $B$ examples from a dataset of size $\check{m}$. In other words, it is equivalent to running DP-SGD for $1/b$ times as many rounds, but with the sampling probability multiplied by $b$.

**Theorem 4.** *Suppose $\mathbf{C}$ is $b$-banded and lower triangular, and the examples participating in each step are chosen according to Alg. 2. Then BANDMF satisfies any standard DP guarantee[6] satisfied by performing $k$ sensitivity-$\kappa$ queries on a dataset of size $\check{m}$ using the Gaussian mechanism, where each query is run on a random subset of examples of size $B$. $\kappa$ is the maximum column norm of $\mathbf{C}$.*

The key idea behind Thm. 4 is the following: assume we have two datasets $D, D'$ that differ in an example in $D_1$ (WLOG), i.e., the differing example can only participate in steps $1, b+1, \ldots, (k-1)b+1$. Then by the banded structure of $\mathbf{C}$ and the standard technique of reducing adaptive queries to non-adaptive queries (see e.g. Claim D.1 in [17]), the first $b$ rows of $\mathbf{C}\mathbf{x}$ (i.e., all the rows where examples in step 1 influence the output) can be viewed as a query on the examples in $D_1$ that were included in step 1, the next $b$ rows can be viewed as an adaptively chosen query on the examples included in steps $b + 1$, and so on. See Fig. 4 for a visualization.

**Generalization to other privacy amplification techniques** Thm. 5 in App. F.3 provides a strong a generalization of Thm. 4. It shows that shuffling, another common amplification technique often used for DP-SGD, can also be applied to our BANDMF. We also provide an explicit algorithm for accounting for amplification via sampling in terms of the `dp_accounting` library [18], along with examples of concrete privacy parameters derived from these corollaries.

**Optimizing the number of bands** Thm. 4 shows that different numbers of bands (with a corresponding sampling scheme) give different privacy amplification guarantees. This implies given a particular privacy (or RMSE) target, one should optimize the number of bands to get the best RMSE (or privacy) possible. This can be done efficiently. Generally, for large values of $\epsilon$ larger numbers of bands perform better, and as $\epsilon \to 0$, eventually amplification dominates so $\hat{b} = 1$ becomes optimal. Details are in App. F.4.

---

[5]and in our case, $k = 1$ always.
[6]Standard DP includes $\epsilon$-DP, $(\epsilon, \delta)$-DP, Rényi DP, zCDP, and Gaussian DP.

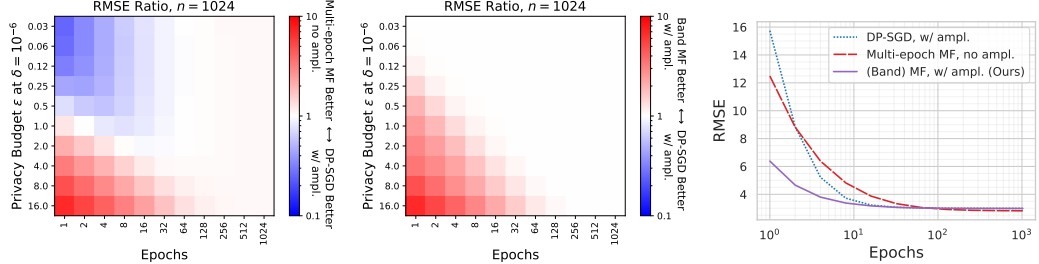

| (a) DP-SGD vs. MULTI-EPOCH MF | (b) DP-SGD vs Ampl. BANDMF | (c) RMSE vs Epochs at $\epsilon = 1$ |

Figure 5: Comparison between DP-SGD, MULTI-EPOCH MF, and BANDMF in terms of RMSE on the prefix-sum queries, for $n = 1024$ iterations, $\epsilon \in \left[\frac{1}{32}, 16\right]$, $\delta = 10^{-6}$, and epochs $\in [1, 1024]$. Color indicates the ratio in RMSE between two mechanisms. Additional details in App. G.

# 6 Experiments

Our experiments on example-level DP for image classification (of CIFAR10) and user-level DP for next word prediction (NWP) (of Stack Overflow NWP) focus on comparing our BANDMF with the existing state-of-the-art MULTI-EPOCH MF [15] and DP-SGD [1]. We finish by showing that BANDMF improves over state-of-the-art [35] for a production mobile keyboard next word prediction model. In all cases, noise $\sigma$ is calibrated using privacy loss distributions [38] to achieve the stated privacy guarantees for zero-out adjacency [35, 48], as implemented in [18]. Following the common convention, amplified results are based on privacy accounting for Poisson sampling, though shuffling was used in non-production training. We find **BANDMF can outperform both mechanisms across a wide range of privacy budgets** to as low as $\epsilon \approx 0.5$. Past this, it is no worse than either.

**RMSE**   We begin by comparing DP-SGD with MULTI-EPOCH MF and BANDMF in terms of their RMSE (Sec. 2) on the prefix workload. This is one measure for how much noise these mechanisms add during training, and is a reasonable proxy for learning performance. Observe in Fig. 5(a) that there are regimes where DP-SGD outperforms MULTI-EPOCH MF and vice versa in terms of RMSE. When then number of epochs equals $n$, DP-SGD reduces to full gradient descent (GD) (and there is no amplification benefit), and the optimal MF mechanism is close to the identity matrix (that is, GD), and so the algorithms become almost identical (MF has a very small advantage, as the identity matrix is not quite optimal for RMSE). However, as shown in Fig. 5(b), we see that BANDMF is always at least as good as DP-SGD in terms of RMSE. The improvement is most potent in the lower number of epochs and higher $\epsilon$ regime, which is standard for large model training. In Fig. 5(c), we see that for fixed $n$ as the number of epochs increases, all mechanisms enjoy improved RMSE, and BANDMF in fact reduces to DP-SGD in that regime (though BANDMF may be outperformed in RMSE by MULTI-EPOCH MF due to our imposed optimization constraint of constant diagonals in $\mathbf{X}$).

**Centralized training with amplification**   Our full experimental setup is described in App. H, and closely follows prior work [15]. We train for 20 epochs on CIFAR-10, and tune all mechanisms to achieve their best performance for each $\epsilon$, using 12 repeated runs. Fig. 1(a) shows that BANDMF with amplification and an optimized number of bands $\hat{b}$ can obtain utility benefits over both prior mechanisms. We find that for $\epsilon \in [2, 5]$, BANDMF achieves a consistent $\approx 1$ percentage point boost in performance over MULTI-EPOCH MF. Below $\epsilon \approx 2$, where DP-SGD previously dominated, we find that BANDMF obtains a benefit around 3 percentage points. These two findings show that BANDMF is able to balance and leverage the benefits of both amplification and correlated noise effectively. As the budget $\epsilon$ gets smaller, we find that BANDMF is equivalent to DP-SGD.

We next consider the now-standard StackOverflow next-word-prediction (NWP) task with user-level differential privacy, again following [15] (full details in App. I), in particular 2052 steps and 6 epochs, with $B = 1000$. The previous state-of-the-art for centralized training at $\epsilon \geq 2$ corresponds to their MULTI-EPOCH MF.[7] We again tune $\hat{b}$ under amplification for optimal RMSE, selecting $\hat{b} = 9, 18, 32, 64$ for $\epsilon = 1, 2, 4, 8$ respectively. At $\epsilon = 16$ we find MULTI-EPOCH MF is optimal. Fig. 1(b) shows substantial improvements for combining amplification with MF for $\epsilon \in [1, 8]$; Table 5 of App. I gives the hyperparameters and accuracy values for this figure.

---

[7]All StackOverflow experiments use the $\hat{b}$=2052 matrices optimized with equal-column-norms via Eq. (6) for MULTI-EPOCH MF, which we observed to be minimally different in terms of RMSE and accuracy.

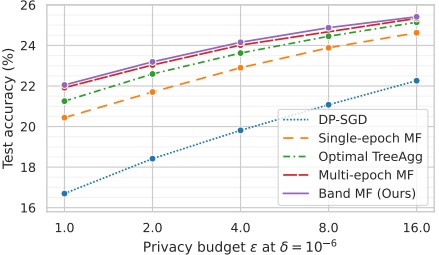
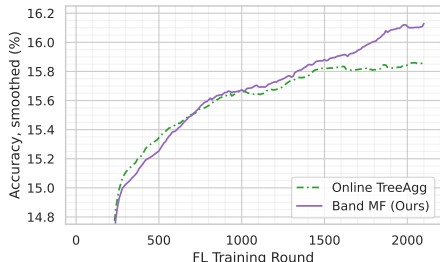

(a) Simulated cross-device FL, SO NWP

(b) Production mobile keyboard training

Figure 6: **BOTH:** Amplification is infeasible as outlined in Sec. 6 and App. K. **LEFT, (A):** Cross-device FL results under $b$=342-min-sep-participation. BANDMF, and MULTI-EPOCH MF (with the application to FL made possible by Thm. 3) outperform all prior work. **RIGHT, (B):** Evaluation accuracy of a language model trained in a real-world FL system. BANDMF achieves higher utility and a stronger $(4.35, 10^{-10})$-DP compared to the $(6.69, 10^{-10})$-DP achieved by ONLINE TREEAGG [35].

**Cross-device federated learning**   We consider SO NWP again, but now assuming each user's data corresponds to the data on one device. We assume 2052 rounds and 6 epochs as above. Amplification is generally not possible in the cross-device setting, and so the prior state-of-the-art was (1) SINGLE-EPOCH MF of Denisov et al. [17] for single-epoch training (using $B = 167$ rather than $B = 1000$), and (2) OPTIMAL TREEAGG, which essentially takes the binary-tree matrix $\mathbf{C}_\mathcal{T}$, and instead of using the less-efficient "online" estimator of Honaker [30], uses the pseudo-inverse $\mathbf{C}_\mathcal{T}^\dagger$ for noise generation (see Sec. 2). The $b$-min-sep-participation sensitivity of $\mathbf{C}_\mathcal{T}$ can still be calculated using the dynamic program of Kairouz et al. [35], while the use of $\mathbf{C}_\mathcal{T}^\dagger$ requires the machinery of Denisov et al. [17]; see e.g. the OPTDECODERHONAKER results of Choquette-Choo et al. [15, Fig. 6]. Our Thm. 3 further enables an upper-bound on the $b$-min-sep-participation sensitivity of the MULTI-EPOCH MF matrices of Choquette-Choo et al. [15]; this incurs a penalty of about 15% (see Table 2 in App. C) compared to $(k, b)$-sensitivity. Fig. 6(a) shows that our BANDMF and MULTI-EPOCH MF again outperform prior baselines (though the previously untested multi-epoch OPTIMAL TREEAGG performs quite well); Table 4 gives the hyperparameters and accuracy values for this figure.

**Application in production cross-device FL**   We fine-tune a Spanish next word prediction model, pretrained on the multilingual C4 dataset [49, 60], with on-device user data using FL. Our setup follows [59], and is described in full in App. K. We compared to an existing implementation of the ONLINE TREEAGG algorithm of Kairouz et al. [35] (not the optimal version using $\mathbf{C}_\mathcal{T}^\dagger$ in simulation). Both algorithms ran for $n = 2000$ training rounds. The BANDMF matrix was optimized for $\hat{b} = 400$ bands; however, the production system only allows approximate control of the separation between participations, and post-hoc we could only bound $b$ by 390 rounds for ONLINE TREEAGG and 385 for BANDMF, necessitating the use of Thm. 3 for the analysis of BANDMF as $b < \hat{b}$.

We used the same clients/round goal of 6500 for both, and *tuned noise multipliers to achieve comparable RMSE*, hence tuning for a stronger privacy guarantee rather than improved accuracy. Fig. 6(b) shows our results, and we see BANDMF actually achieves a slight improvement in accuracy, possibly due to learning-rate cooldown (which was only implemented for BANDMF). **Our primary result is then that we are able to improve the privacy guarantee from $\rho$=0.52-zCDP for ONLINE TREEAGG to $\rho$=0.24-zCDP for BANDMF,** or $(\epsilon=6.69, \delta=10^{-10})$-DP to $(\epsilon=4.35, \delta=10^{-10})$-DP. Details of the privacy guarantee following the best practices of Ponomareva et al. [48] are in App. K.1.

## 7   Discussion and Limitations

In this paper, we proposed the BANDMF mechanism, which extends MF-DP-FTRL and enjoys the benefits of privacy amplification. This allows it to *solely operate above* the previous Pareto frontier defined by both amplified DP-SGD and MF-DP-FTRL in centralized training scenarios. Moreover, BANDMF is well-suited to federated training scenarios, and improves state-of-the-art there as well. Additionally, the computational overhead of BANDMF is less than MF-DP-FTRL by a factor of $b/n$. It still has a $b\times$ time and space overhead compared to DP-SGD, which can be prohibitive for very large models with billions of parameters. This is an interesting and important future research direction.

**Acknowledgement**

The authors thank the early feedback and discussion from Natalia Ponomareva, and the support of FL production training from Yanxiang Zhang and Yuanbo Zhang.

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

# A  Notation summary

| | |
|---|---|
| $n$ | Number of steps of the streaming linear query (SGD steps or FL rounds) |
| $m$ | Total number of records (examples or users) in the database/dataset |
| $b$ | Minimum separation between participations; $b = 1$ allows participation in every step |
| $\hat{b}$-banded matrix | A (general) matrix $\mathbf{X}$ is $\hat{b}$-banded if for all $i, j \in [n]$, $|i - j| \geq \hat{b}$ implies $\mathbf{X}_{[i,j]} = 0$. We use $b$ to refer to min-separation, and for example write $b$-banded when we set the number of bands equal to the min-separation; we use $\hat{b}$-banded for a number of bands possibly different than $b$. |
| $k$ | The maximum number of times any user might participate in training |
| $d$ | Dimension of per-step user contributions (e.g., model size) |
| $\mathbf{x}_i \in \mathbb{R}$ or $\mathbb{R}^d$ | Sum of per-example gradients (or per-user model updates) on step $i$ |
| $\mathbf{x} \in \mathbb{R}^{n \times d}$ | Stream of inputs $\mathbf{x}_i$, equiv. matrix with rows $\mathbf{x}_i$ (so $\mathbf{x}_i = \mathbf{x}_{[i,:]}$) |
| $\zeta$ | Clipping norm that limits the size of per-example contributions to $\mathbf{x}_i$ |
| $\pi \subseteq [n]$ | Participation pattern, the set of steps that an example participates in |
| $\Pi$ | Participation schema, set of sets of steps (set of all $\pi$) an example could participate in |
| $\mathfrak{D}$ | $= \{\mathbf{x} - \tilde{\mathbf{x}} \mid (\mathbf{x}, \tilde{\mathbf{x}}) \in \mathbf{N}\}$, the set of deltas between neighboring input streams $\mathbf{x}, \tilde{\mathbf{x}}$. |
| $\mathcal{D}$ | Corners of $\mathfrak{D}$ when assumed to be a polytope, $\mathfrak{D} = \text{conv}(\mathcal{D})$. |
| $(k, b)$-participation | participation schema $\Pi$ with at most $k$ participations, separated by exactly $b$ steps |
| $b$-min-sep-participation | Relaxation of of $(k, b)$-participation where participations have separation at least $b$ |
| $\mathbf{A} \in \mathbb{R}^{n \times n}$ | Lower-triangular linear query matrix to be factorized as $\mathbf{A} = \mathbf{BC}$ |
| $\mathbf{M}^\dagger$ | Moore-Penrose pseudoinverse of matrix $\mathbf{M}$ |
| $\mathbf{M}^\top$ | Transpose of $\mathbf{M}$ |
| $\mathbf{M}_{[i,j]}$ | The $(i, j)^{\text{th}}$ entry of matrix $\mathbf{A}$ |
| $\mathbf{M}_{[i,:]}$ and $\mathbf{M}_{[:,j]}$ | The $i^{\text{th}}$ row and $j^{\text{th}}$ column of $\mathbf{M}$ (numpy-style indexing) |
| $\text{conv}(S)$ | Convex hull of the set $S$ |
| $[n]$ | $= \{1, \dots, n\}$ |
| $\|\mathbf{X}\|_F$ | The Frobenius norm of a matrix $\mathbf{X}$ |

Table 1: Summary of notation

## A.1  Intuition behind MF-DP-FTRL

Fig. 7 compares DP-SGD and MF-DP-FTRL. To gain an intuition for why MF-DP-FTRL can perform better than DP-SGD, observe that vanilla SGD has iterates $\theta_t = \theta_0 - \eta \sum_{i=1}^{t} \hat{\mathbf{x}}_i$, and hence when the noisy gradients $\hat{\mathbf{x}}_i$ are added, the $\mathbf{C}_{[i,j]}^{-1} \mathbf{z}_{[i,:]}$ terms in MF-DP-FTRL serve to *cancel out* some of the noise introduced on previous rounds. The noise cancellation reduces the total error in all the prefix sums of gradients $\sum_{i=1}^{t} \hat{\mathbf{x}}_i$ for $t \in [n]$, but also worsens the privacy guarantee of the mechanism, i.e. increases its sensitivity. The privacy worsens as e.g., an adversary trying to learn $\mathbf{x}_1$ via $\hat{\mathbf{x}}_1$ can partially learn the value of $\mathbf{z}_1$ from $\hat{\mathbf{x}}_2$, whereas in DP-SGD $\hat{\mathbf{x}}_1$ and $\hat{\mathbf{x}}_2$ are uncorrelated. Hence there is a tradeoff between the total error and sensitivity (see next paragraph): DP-SGD sets $\mathbf{C}_{[i,j]}^{-1} = 0$

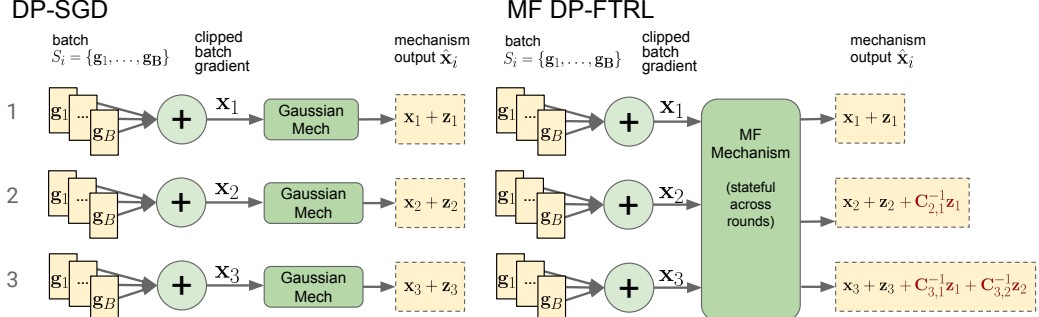

Figure 7: **MF-DP-FTRL (Alg. 1) enables noise cancelling across steps, where DP-SGD does not**. The entries $\mathbf{C}_{i,j}^{-1}$ are mostly negative (in $[0, -1)$) in matrices $\mathbf{C}^{-1}$ we consider (see Fig. 8). Thus, the red terms show that MF-DP-FTRL "cancels out" noise added on earlier iterations. For simplicity, we assume $\mathbf{C}^{-1}$ has 1s on the main diagonal and entries $\mathbf{C}_{i,j}^{-1}$ otherwise, with $\mathbf{z}_i := \mathbf{z}_{[i,:]}$ the rows of $\mathbf{z}$.

below the main diagonal, effectively minimizing sensitivity (assuming a fixed normalization of the main diagonal), but with a large total error due to no noise cancellation. On the other hand, MF-DP-FTRL can arrive at a better compromise between mechanism sensitivity and the total error. This is formalized in the optimization problem of Sec. 4.

Without a sampling assumption, the implied DP adversary knows which examples participated in batch $S_i$, and for DP-SGD with uncorrelated noise, knows they only need to "attack" $\hat{\mathbf{x}}_i$. However, with MF-DP-FTRL, the information from $S_i$ can potentially be masked with a larger amount of initial noise in $\hat{\mathbf{x}}_i$, which is then canceled out over subsequent rounds. "Spreading out" the release of information about batch $S_i$ over a larger number of iterations in this way can intuitively provide better privacy, while still allowing for accurate partial sums of gradients (and hence SGD iterates). This is, in a loose sense, similar to the way SGD with sampling "hides" information about a particular example at randomly chosen iterations.

## B  Dropping the $\operatorname{diag}(\mathbf{X}) = \mathbf{1}$ constraint

As discussed in Sec. 4, BANDMF by default imposes an equal column norm constraint on the generated factorization. In the optimization problem, this is accomplished by imposing the constraint $\operatorname{diag}(\mathbf{X}) = \mathbf{1}$. In this section we show how we can solve the optimization problem without this constraint for $(k, b)$-participation. This optimization problem is formulated to minimize total squared error with respect to $(k, b)$-participation, although in principle the optimized matrices could be used in the $b$-min-sep-participation setting with some degradation in solution quality. Prop. B.1 provides an expression for efficiently computing the sensitivity of a $b$-banded matrix.

**Proposition B.1.** *Let* $\mathbf{X} \in \mathbf{S}_+^n$ *be a* $b$-*banded matrix, and let* $\Pi$ *denote the* $(k, b)$-*participation schema. Then*

$$\operatorname{sens}_\Pi(\mathbf{X}) = \max_{i=1,\dots,b} \sum_{j=0}^{k-1} \operatorname{diag}(\mathbf{X})_{i+jb}.$$

To integrate this expression into the optimization problem, we can replace the $\operatorname{diag}(\mathbf{X}) = \mathbf{1}$ constraint with $b$ linear constraints on $\operatorname{diag}(\mathbf{X})$. This modification does not affect the convexity of the problem, although it does slightly complicate the algorithms needed to solve it. To handle this new problem, our approach is to replace the gradient with respect to $\mathbf{X}$ at each iteration, with a *projected gradient*, which is obtained by setting $v_i = \sum_{j=0}^{k-1} \operatorname{diag}(\Delta\mathbf{X})_{i+jb}$ for all $i = 1, \dots, b$, and setting $\operatorname{diag}(\Delta\mathbf{X})_{i+jb} = \operatorname{diag}(\Delta\mathbf{X})_{i+jb} - v_i/k$. This ensures that sensitivity does not change between iterations of the numerical optimization procedure.

For the reasons mentioned in Sec. 4, by default we impose the simpler constraint $\operatorname{diag}(\mathbf{X}) = \mathbf{1}$. In App. C, we provide some numerical comparisons between these two approaches. Specifically, the rows of Table 2 with *Equal column norms?* (F)alse correspond to the approach described here; observe this results in slightly improved RMSE under $(k, b)$-participation compared to the corresponding rows with *Equal column norms?* (T)rue.

| Mechanism | Matrix | | Sensitivity | | | Error | |
| | Bands $\hat{b}$ | Equal column norms? (Ours) | $k{=}1$ [17] | $(k{=}6, b{=}342)$ [15] | $b{\geq}342$-min-sep (Ours) | (A) RMSE under $(k{=}6, b{=}342)$ [15] | (B) RMSE under $b{\geq}342$-min-sep (Ours) |
|---|---|---|---|---|---|---|---|
| OPTIMAL TREEAGG [30, 35] | - | F | 0.32 | 1.00 | 1.00 | 1.53 | 1.53 |
| DP-SGD [1] | 1 | T | 0.41 | 1.00 | 1.00 | 9.63 | 9.63 |
| MF ($b{=}128$) (Ours) | 128 | F | 0.52 | 1.00 | 1.04 | 1.23 | 1.29 |
| MF ($b{=}128$) (Ours) | 128 | T | 0.41 | 1.00 | 1.00 | 1.27 | 1.27 |
| MF ($b{=}342$) (Ours) | 342 | F | 0.52 | 1.00 | 1.04 | 1.04 | 1.08 |
| MF ($b{=}342$) (Ours) | 342 | T | 0.41 | 1.00 | 1.00 | 1.05 | **1.05** |
| MF [15] | - | F | 0.50 | 1.00 | ≤1.15 | **1.00** | 1.15 |
| MF [15] | - | T | 0.41 | 1.00 | ≤1.13 | 1.01 | 1.14 |

Table 2: A comparison of matrix mechanisms for $n = 2052$ under different participation patterns. **Banded matrices are near-optimal under $(k, b)$-participation and best under $b$-min-sep-participation.** Each error is computed under the indicated measure of sensitivity. Thus, the error in column (B) can be obtained by multiplying the error in column (A) by the corresponding entry under $b{\geq}342$ sensitivity.

## C   Empirical evaluation of banded matrices

Table 2 compares the matrix mechanisms studied under different participation patterns but normalized to have sensitivity $\mathrm{sens}(\mathbf{C}) = 1$ under $(k{=}6, b{=}342)$-participation. The sensitivity under single participation $k = 1$ is lowest as expected. With column normalization, sensitivity is also 1 under $b{\geq}342$-min-sep-participation. We make the following observations:

- For the MF mechanisms, column normalization hurts RMSE for $(k, b)$-participation compared to the approach of App. B (as it is an additional constraint), but actually improves RMSE under $b$-min-sep-participation.

- We conjecture that the $(k, b)$-participation optimized matrices (MF without column normalization, App. B) are optimal for the prefix-sum workload[8]; With this in mind, we see there is at most a small increase in RMSE for switching to the more challenging $b$-min-sep-participation schema ($1.00 \to 1.05$) . If (as we further conjecture) the optimal matrices for prefix-sum in fact are $k$-banded, the gap is even smaller (at most $1.04 \to 1.05$). Hence, at least for the prefix-sum workload $\mathbf{A}$, there is limited room for improvement in developing optimization procedures that directly optimize over the larger feasible set offered by $b$-min-sep-participation.

- Using fewer than $b$ bands does degrade performance on the RMSE metric, with DP-SGD being the extreme case, yielding prefix sum estimates almost $10\times$ worse than the MF mechanisms.

- The results of Denisov et al. [17] imply that the binary-tree $\mathbf{C}$ matrix can in fact be used in the online setting, with the Moore-Penrose pseudo-inverse giving the optimal decoder for RMSE [15], corresponding to the 'full' estimator of Honaker [30]. We include this in the table as a baseline, and see that it is in general outperformed by our MF mechanisms by about $1.5\times$ in RMSE.

---

[8]This conjecture is not trivially true, as we still enforce a non-negativity or orthogonality constraint; see Choquette-Choo et al. [15, Appendix I.3]. Hence the conjecture is that these constraints are already satisfied by the optimal matrix for this workload.

# D    Example structures of MF

Figs. 8 and 9 show the structure of some of the key matrix factorization approaches considered in this work. One can immediately see the impact of the $(k, b)$-participation schema in the optimal matrices, in particular for the non-banded MULTI-EPOCH MF matrices (the two top-right matrices), where $\mathbf{C}$ contains diagonals of negative entries separated by $b$ steps. In the bottom two rows, we see that requiring equal column norms ("EN-" for equal norms) has a relatively minor impact on the structure of the matrices.

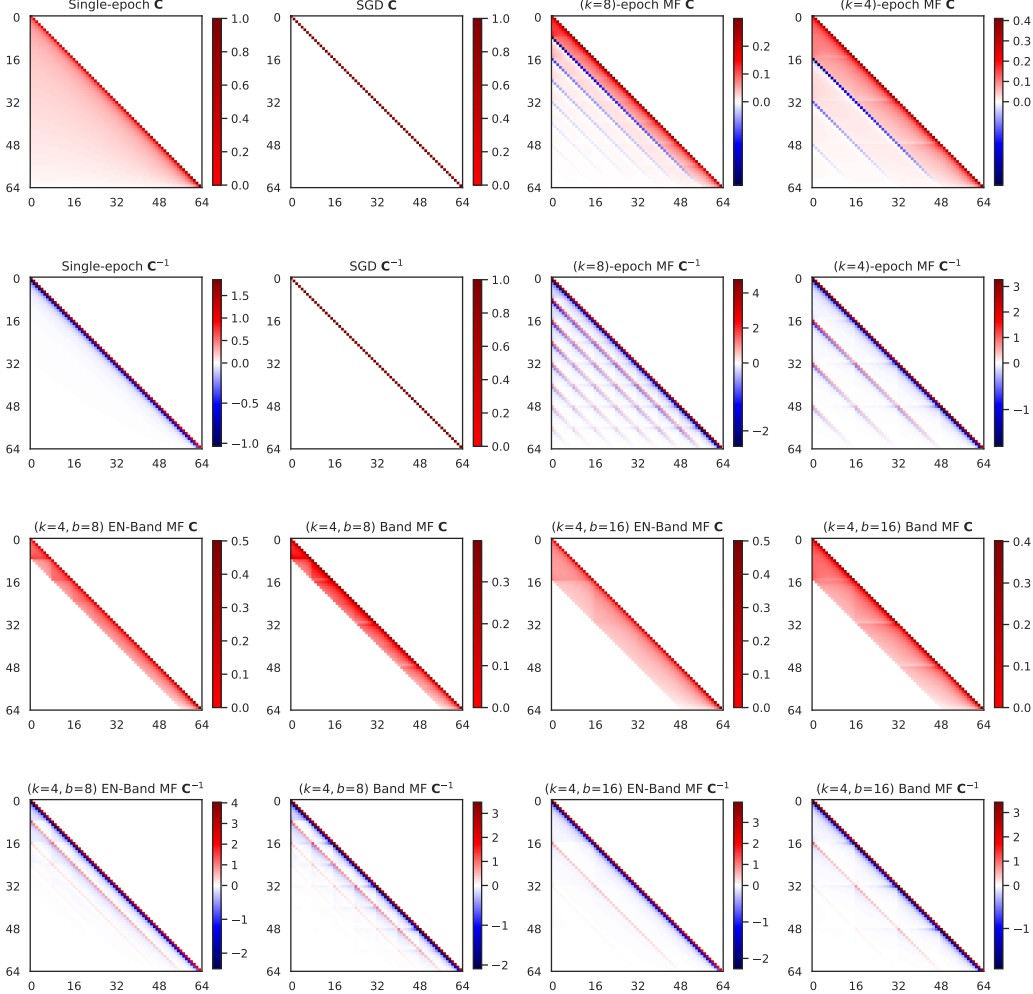

Figure 8: Factorizations for $n = 64$ of the prefix-sum workload ($\mathbf{A}$ taken to be the lower-triangular matrix of 1s). For each factorization $\mathbf{A} = \mathbf{BC}$, we show $\mathbf{C}$ and its inverse $\mathbf{C}^{-1}$, as the inverse is the matrix used in noise generation. Single-epoch is the approach of Denisov et al. [17], SGD is simply the identity matrix $\mathbf{I}$ (shown for completeness), and $(k{=}8)$-epoch MF and $(k{=}4)$-epoch are the MULTI-EPOCH MF approach of Choquette-Choo et al. [15] for 8 and 4 epochs, respectively. For our banded matrices (3rd and 4th rows), we fix 4 epochs ($b = 16$), and show $\hat{b}{=}8$ and $\hat{b}{=}16$ bands, with column normalization ("EN-") and without.

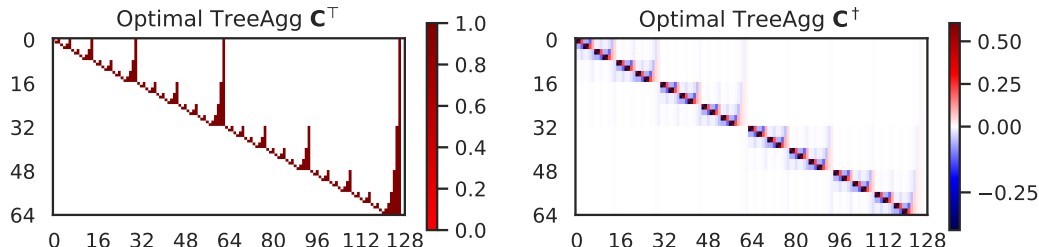

Figure 9: The transpose of binary-tree encoder matrix $\mathbf{C}_{\mathcal{T}}$, and its pseudoinverse $\mathbf{C}_{\mathcal{T}}^{\dagger}$, which corresponds to the "full" or optimal decoder of Honaker [30]. This is the matrix used in OPTIMAL TREEAGG in Fig. 6[a].

# E    Algorithms and Analysis for Sec. 3

## E.1    Algorithms

---

**Algorithm 3** (VECSENS): Maximum of $\langle \mathbf{v}, \mathbf{u} \rangle$ where $\mathbf{u}$ is a vector in the $\ell_\infty$ unit ball satisfying $\Pi_b$.

---

**Inputs:** min-separation $b$, vector $\mathbf{v}$, max participations $k$
Initialize $F \in \mathbb{R}^{n \times k}$
**for** $m = 1, \ldots, k$ **do**
    **for** $i = n, \ldots, 1$ **do**       $\triangleright$ We use the convention that $F[s, t] = 0$ if $s, t$ are out-of-bounds.
$$F[i, m] = \max\left(\mathbf{v}_i + F[i + b, m - 1], F[i + 1, m]\right)$$
**return** $F[1, k]$

---

---

**Algorithm 4** Efficient sensitivity upper bound for $b$-min-sep-participation

---

**Inputs:** min-separation $b$, matrix $\mathbf{X}$, max participations $k$
Initialize $F \in \mathbb{R}^{n \times k}$, $\mathbf{v} \in \mathbb{R}^n$.
**for** $j = 1, \ldots, n$ **do**
    $\mathbf{v}_i = \text{VECSENS}(b, |\mathbf{X}_{[i,:]}|, k)$
**return** $\sqrt{\text{VECSENS}(b, \mathbf{v}, k)}$

---

---

**Algorithm 5** Efficient sensitivity calculation for $b$-min-sep-participation, assuming $\mathbf{X}$ is $b$-banded.

---

**Inputs:** min-separation $b$, $b$-banded matrix $\mathbf{X}$, max participations $k$.
**return** $\sqrt{\text{VECSENS}(b, \text{diag}(\mathbf{X}), k)}$

---

## E.2    Analysis

**Proposition E.1.** *The sensitivity of* $\mathbf{C}$ *for a given participation schema* $\Pi$ *may be expressed as:*

$$\text{sens}_\Pi(\mathbf{C})^2 = \max_{\pi \in \Pi} \sup_{\mathbf{u} \in \mathfrak{D}} \text{tr}\left(\left[\mathbf{P}_\pi \mathbf{C}^\top \mathbf{C} \mathbf{P}_\pi\right]\left[\mathbf{u}\mathbf{u}^\top\right]\right), \tag{7}$$

*where* $\mathbf{P}_\pi$ *represent the axis-aligned projection onto the set of rows indexed by* $\pi$; *that is,* $\mathbf{P}_\pi[i, i] = 1$ *for* $i \in \pi$, *and* $0$ *otherwise. Assuming that* $\mathfrak{D}$ *represents a set of matrices with rows bounded by* $\ell_2$ *norm* $1$, *this can be upper bounded by:*

$$\max_{\pi \in \Pi} \sum_{i,j \in \pi} |\mathbf{X}_{[i,j]}|.$$

*where* $\mathbf{X} = \mathbf{C}^\top \mathbf{C}$. *This upper bound is tight when* $\mathbf{P}_\pi \mathbf{C}^\top \mathbf{C} \mathbf{P}_\pi \geq 0 \forall \pi \in \Pi$, *and is independent of the dimension* $d$ *of the rows of* $\mathbf{u}$.

*Proof.* Recall that $\Pi$ determines the rows of $\mathbf{u}$ which may be nonzero in the definition Eq. (2). Take some $\mathbf{u} \in \mathfrak{D}$, an element of $\mathbb{R}^{n \times d}$, which therefore has nonzero rows only at some set of indices $\pi \in \Pi$. Note, clearly $\mathbf{u} = \mathbf{P}_\pi \mathbf{u}$, $\mathbf{P}_\pi^\top = \mathbf{P}_\pi$, and $\mathbf{P}_\pi = \mathbf{P}_\pi \mathbf{P}_\pi$.

Therefore

$$
\begin{aligned}
\|\mathbf{C}\mathbf{u}\|_F^2 &= \operatorname{tr}\left([\mathbf{C}\mathbf{P}_\pi \mathbf{u}]^\top \mathbf{C}\mathbf{P}_\pi \mathbf{u}\right) = \operatorname{tr}\left(\mathbf{u}^\top \mathbf{P}_\pi^\top \mathbf{C}^\top \mathbf{C}\mathbf{P}_\pi \mathbf{u}\right) \\
&= \operatorname{tr}\left(\mathbf{P}_\pi \mathbf{P}_\pi \mathbf{C}^\top \mathbf{C}\mathbf{P}_\pi \mathbf{P}_\pi \mathbf{u}\mathbf{u}^\top\right) = \operatorname{tr}\left(\left[\mathbf{P}_\pi \mathbf{C}^\top \mathbf{C}\mathbf{P}_\pi\right]\left[\mathbf{P}_\pi \mathbf{u}\mathbf{u}^\top \mathbf{P}_\pi\right]\right) \\
&= \operatorname{tr}\left(\left[\mathbf{P}_\pi \mathbf{C}^\top \mathbf{C}\mathbf{P}_\pi\right]\left[\mathbf{u}\mathbf{u}^\top\right]\right).
\end{aligned}
\tag{8}
$$

This implies the statement Eq. (7) by the definition of sensitivity and neighboring in our setting.

Now, let $\mathbf{X}_\pi \coloneqq \mathbf{P}_\pi \mathbf{C}^\top \mathbf{C}\mathbf{P}_\pi$ be the matrix formed by zeroing out the rows and columns *not* indexed by $\pi$ from $\mathbf{X}$. Assume that every $\mathbf{u} \in \mathfrak{D}$ has row norms bounded by 1. Expanding the trace in Eq. (7), writing $x_{ij}$ for the elements of $\mathbf{X}_\pi$ and $\mathbf{u}_{[j,:]}$ for the $j^{th}$ row of $\mathbf{u}$, we have

$$
\operatorname{tr}\left(\mathbf{X}_\pi \mathbf{u}\mathbf{u}^\top\right) = \sum_{i=1}^k \sum_{j=1}^k x_{ij}\langle \mathbf{u}_{[i,:]}, \mathbf{u}_{[j,:]}\rangle \le \sum_{i=1}^k \sum_{j=1}^k |x_{ij}|
$$

which yields the claimed bound. When $\mathbf{X}_\pi$ is elementwise nonnegative, taking $\mathbf{u}_{[i,:]} = \mathbf{u}_{[j,:]}$ for any unit vector shows the claimed tightness in this case.

$\square$

**Remark.** This statement can be viewed as a partial extension of [15, Theorem G.1]. It does not imply every case handled there, but also implies results which cannot be derived from that Theorem.

*Proof of Thm. 2.* Conclusion (1) is implied by (2), noting that the conditions on $\mathbf{C}$ imply that Alg. 5 will return a value at most $\kappa\sqrt{k'}$ in this setting.

For (2), let $c \in \mathbb{R}^n$ with entries $c_i = \|\mathbf{C}_{[:,i]}\|^2$ for $i \in \{0, \dots, n-1\}$. We have

$$
\operatorname{sens}_\Pi^1(\mathbf{C}) = \max_{\pi \in \Pi_b} \|\mathbf{C}\mathbf{u}(\pi)\| = \max_{\pi \in \Pi_b} \left\|\sum_{i \in \pi} \mathbf{C}_{[:,i]}\right\| = \max_{\pi \in \Pi_b} \sqrt{\sum_{i \in \pi} c_i}
\tag{9}
$$

where $\mathbf{u}(\pi) \in \{0,1\}^n$ is given by $\mathbf{u}(\pi)_i = 1$ if $i \in \pi$ and $0$ otherwise. The last equality follows from the orthogonality condition on sufficiently separated columns of $\mathbf{C}$ trivially implied by bandedness. It is straightforward to verify the dynamic program of Alg. 3 constructs a feasible $\pi$ which attains the maximum. $\square$

*Proof of Thm. 3.* Via Prop. E.1, the result follows from showing that Alg. 4 outputs a value at least as large as $\sum_{(i,j)\in\pi} |\mathbf{X}_{ij}|$ for any $\pi \in \Pi_b$. So let $\hat{\pi}$ be an element of $\Pi_b$. Note that VECSENS is monotonically increasing in values of the vector $\mathbf{v}$ if $\mathbf{v}$ is nonnegative, and therefore Alg. 4 is monotonically increasing in absolute values of $\mathbf{X}$. Therefore we will have our conclusion (3) if we can show that, for $\mathbf{X}_{\hat{\pi}}$ the matrix formed by zeroing out all rows and columns of $\mathbf{X}$ not indexed by $\hat{\pi}$, Alg. 4 returns the value $\sum_{(i,j)\in\pi} |\mathbf{X}_{ij}|$. Yet this is straightforward by the characterization of VECSENS as an oracle for computing the maximum of $\langle \mathbf{v}, \mathbf{u}\rangle$, where $\mathbf{u}$ is a vector in the $\ell_\infty$ unit ball. $\square$

*Proof of Prop. 4.1.* The proof will be constructive. Let $\mathbf{J}$ be the $n \times n$ *exchange matrix* defined as

$$
\mathbf{J} = \begin{bmatrix} & & & 1 \\ & & 1 & \\ & \iddots & & \\ 1 & & & \end{bmatrix}
$$

Let $\mathbf{Y} = \mathbf{J}\mathbf{X}\mathbf{J}$ and note that $\mathbf{Y}$ is symmetric and positive definite. Let $\mathbf{H} = \text{Cholesky}(\mathbf{Y})^\top$ and note that (1) $\mathbf{H}^\top \mathbf{H} = \mathbf{Y}$ by definition of Cholesky decomposition, (2) $\mathbf{H}$ is upper triangular, and (3) $\mathbf{H}$ is $\hat{b}$-banded by Du Croz et al. [19].

We will show that for $\mathbf{C} = \mathbf{JHJ}$, we have (1) $\mathbf{X} = \mathbf{C}^\top\mathbf{C}$, (2) $\mathbf{C}$ is lower triangular, and (3) $\mathbf{C}$ is $\hat{b}$-banded.

For Claim (1) observe that:

$$\begin{aligned}
\mathbf{C}^\top\mathbf{C} &= (\mathbf{JHJ})^\top(\mathbf{JHJ}) \\
&= \mathbf{J}^\top\mathbf{H}^\top\mathbf{J}^\top\mathbf{JHJ} \\
&= \mathbf{J}(\mathbf{H}^\top\mathbf{H})\mathbf{J} \\
&= \mathbf{JYJ} \\
&= \mathbf{JJXJJ} \\
&= \mathbf{X}
\end{aligned}$$

For Claim (2) and (3), note that left-multiplying by $\mathbf{J}$ reverses the rows and right-multiplying by $\mathbf{J}$ reverses the columns, and therefore $\mathbf{C}_{[i,j]} = \mathbf{H}_{n-i+1,n-j+1}$.

For Claim (2), we need to show $\mathbf{C}_{[i,j]} = 0$ if $i < j$. If $i < j$ then $n - i + 1 > n - j + 1$, and since $\mathbf{H}$ is upper triangular, we know $\mathbf{H}_{[n-i+1,n-j+1]} = 0$, as desired.

For Claim (3), we need to show that $\mathbf{C}_{[i,j]} = 0 \; if |i-j| \geq \hat{b}$. Observe that if $|(n-i+1)-(n-j+1)| = |i-j|$ and therefore since $\mathbf{H}$ is $\hat{b}$-banded, so is $\mathbf{C}$.

This completes the proof. $\qquad\square$

# F   Additional Analysis for Sec. 5

In this section we prove our general amplification statement Theorem 5, of which Theorem 4 is a corollary. Recall that we use $b$ instead of $\hat{b}$ in this appendix since our sampling scheme enforces $(k, b)$-participation. Throughout this section, we slightly abuse notation by letting $i \pmod b = b$ instead of 0 if $i/b$ is integer.

## F.1   Algorithms for Sampling

We first give the general sampling scheme (Alg. 6) as well as sequence of queries (Alg. 7) that provides an upper bound on the privacy guarantees of DP-MF using this sampling scheme.

---

**Algorithm 6** Sampling scheme for banded DP-MF

---

**Inputs:** Dataset $D$, sampling distribution $\mathcal{S}$ over $(2^{[\breve{m}]})^k$, noise standard deviation $\sigma$.
$D_1, \ldots, D_b \leftarrow$ arbitrary partition of $D$ such that $\forall j : |D_j| = \breve{m}$.
Let $D_j = \{d_{j,1}, d_{j,2}, \ldots, d_{j,\breve{m}}\}$ for each $j$.
**for** $j = 1, 2, \ldots, b$ **do**
    Sample $k$ sets to index $D_j$ as $(S_j, S_{b+j}, \ldots, S_{(k-1)b+j}) \sim \mathcal{S}$, with $S_j \subseteq [\breve{m}]$.
**for** $i = 1, 2, \ldots, n$ **do**
    Let $j = i \pmod b$; compute $\mathbf{x}_i$ by querying $\{d_{j,\ell} : \ell \in S_i\}$.
Let $\mathbf{x} = [\mathbf{x}_1, \ldots, \mathbf{x}_n]^\top \in \mathbb{R}^{n \times d}$, release $\mathbf{Cx} + \mathbf{z}$ with each entry of $\mathbf{z}_{[i,j]} \sim \mathcal{N}(0, \sigma^2)$.
$\triangleright$ If $\mathbf{C}$ is lower-triangular, results can also be released in streaming fashion

---

---

**Algorithm 7** Sequence of queries that bounds privacy of Alg. 6

---

**Inputs:** Dataset $\tilde{D} = \{d_1, d_2, \ldots, d_{\breve{m}}\}$, sampling distribution $\mathcal{S}$ over $(2^{[\breve{m}]})^k$.
Sample $(S_1, S_2, \ldots, S_k) \sim \mathcal{S}$.
**for** $i = 1, 2, \ldots, k$ **do**
    $\tilde{D}_i \leftarrow \{d_j : j \in S_i\}$.
    Perform (adaptively chosen) sensitivity $\Delta$ query on $\tilde{D}_i$ with noise $\mathcal{N}(0, \sigma^2)$.

---

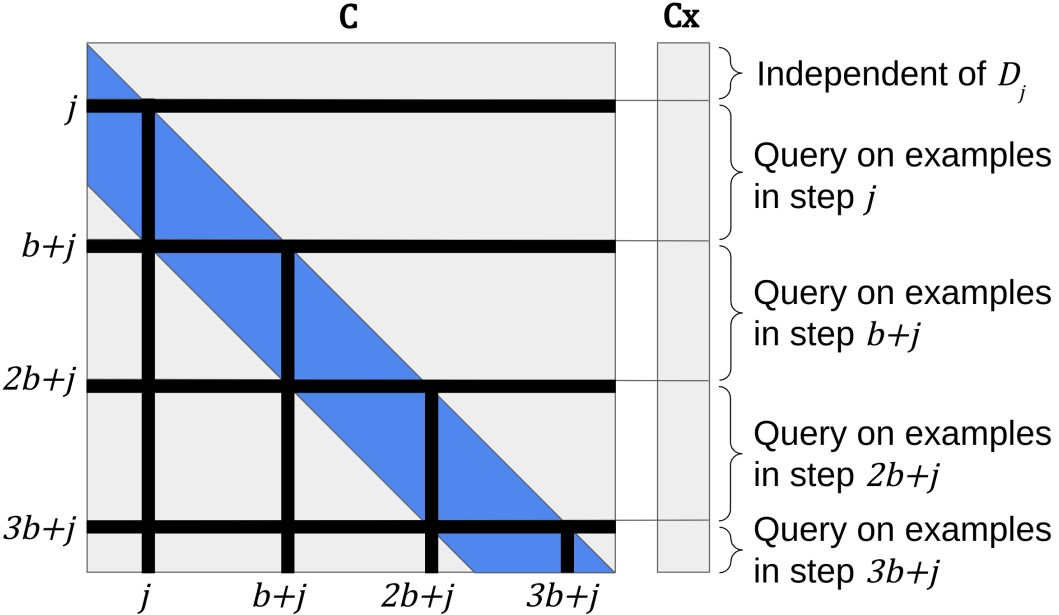

Figure 10: A visualization of how we can decompose a banded matrix mechanism into independent queries on $D_j$ (as in Alg. 7) under our sampling scheme.

## F.2 General Amplification Statement and Proof

Given the sampling scheme and query sequence, we can now state our general amplification statement:

**Theorem 5.** *Suppose* $\mathbf{C}$ *is $b$-banded and lower triangular, and the examples participating in each step are chosen according to Alg. 6 with a given choice of $\mathcal{S}$. Then* BANDMF *satisfies any standard DP guarantee[9] satisfied by Alg. 7 in App. F.1 with* $\kappa = \max_{i \in [n]} \|\mathbf{C}\mathbf{e}_i\|_2 = \max_{i \in [n]} \sqrt{\mathbf{X}_{i,i}}$ *and the same choice of $\mathcal{S}$.*

*Proof.* Consider two datasets $D, D'$ that differ by an example contained in the partition subset $D_j$. We argue about the privacy of $\mathbf{C}\mathbf{x} + \mathbf{z}$. For simplicity we assume $j$ is such that $(k-1)b + j \le n$; elements in $D_j$ such that $j$ does not satisfy this condition can potentially participate $k - 1$ times instead of $k$, and in turn the privacy guarantee we can prove for these elements can only be stronger.

Since $\mathbf{C}$ is $b$-banded, we can partition the rows of $\mathbf{C}$ into $k + 1$ subsets

$$R_j, R_{b+j}, R_{2b+j} \ldots R_{(k-1)b+j}, R_\varnothing,$$

where $R_j$ (resp. $R_{b+j}, R_{2b+j} \ldots R_{(k-1)b+j}$) denotes the set of rows in $\mathbf{C}$ for which the $j$th entry is non-zero, and $R_\varnothing = [n] \smallsetminus (R_j \cup R_{b+j} \cup \ldots)$, i.e., $R_\varnothing$ are the rows not included in any of these sets, i.e., rows of $\mathbf{C}$ where entries $j, b + j, \ldots$ are all zero. The fact that $\mathbf{C}$ is lower-triangular and $b$-banded ensures that these subsets do not overlap, i.e., this is a valid partition as can be observed in Fig. 4.

Let $\mathbf{C}_R$ denote $\mathbf{C}$ restricted to the set of rows in $R$. From the perspective of an adversary distinguishing $D$ from $D'$, each row of $(\mathbf{C}\mathbf{x} + \mathbf{z})_{R_\varnothing} = \mathbf{C}_{R_\varnothing}\mathbf{x} + \mathbf{z}_{R_\varnothing}$ has a distribution independent of whether $D$ or $D'$ was used. So it suffices to give privacy guarantees for outputting only $(\mathbf{C}\mathbf{x} + \mathbf{z})_{R_j}, (\mathbf{C}\mathbf{x} + \mathbf{z})_{R_{b+j}}, \ldots, (\mathbf{C}\mathbf{x} + \mathbf{z})_{R_{(k-1)b+j}}$.

We can decompose rows $R_j$ of $\mathbf{C}\mathbf{x} + \mathbf{z}$ as follows:

$$(\mathbf{C}\mathbf{x} + \mathbf{z})_{R_j} = \mathbf{C}_{R_j}\mathbf{x} + \mathbf{z}_{R_j} = \mathbf{C}_{R_j}\mathbf{x}_j + \mathbf{C}_{R_j}\mathbf{x}_{-j} + \mathbf{z}_{R_j}. \tag{10}$$

Where $\mathbf{x}_j$ denotes $\mathbf{x}$ with all rows except $j$ zeroed out, and $\mathbf{x}_{-j}$ denotes $\mathbf{x} - \mathbf{x}_j$, i.e., $\mathbf{x}$ with row $j$ zeroed out. By the $b$-banded property of $\mathbf{C}$, $\mathbf{C}_{R_j}\mathbf{x}_{-j}$ has 0 sensitivity to the examples in $D \smallsetminus D_j$. Then,

---

[9]Standard DP includes $\epsilon$-DP, $(\epsilon, \delta)$-DP, Rényi DP, zCDP, and Gaussian DP.

by Eq. (10), for $i \in R_j$, we observe that the $i$th row of $(\mathbf{Cx} + \mathbf{z})_{R_j}$ corresponds to an (adaptive) query made with $\ell_2$-sensitivity $\mathbf{e}_i^\top \mathbf{Ce}_j$ to the examples used in step $j$, i.e., those given by $D_j$ and $S_j$, and noise $N(0, \sigma^2)^d$. So $(\mathbf{Cx} + \mathbf{z})_{R_j}$ corresponds to a sequence of adaptive queries on the examples used in step $j$, and answering this sequence of queries satisfies any standard privacy guarantee satisfied by answering a single (scalar, adaptively chosen) query with sensitivity $\|\mathbf{Ce}_j\|_2$ to the example chosen in step $j$ and noise $N(0, \sigma^2)$ by Claim D.1 in [17].

The same logic applies to each of $(\mathbf{Cx} + \mathbf{z})_{R_{b+j}}, \ldots, (\mathbf{Cx} + \mathbf{z})_{R_{(k-1)b+j}}$. Putting it all together and taking a max over the sensitivity of the individual queries, releasing $\mathbf{Cx} + \mathbf{z}$ satisfies any standard privacy guarantee satisfied by answering $k$ adaptively chosen queries, with sensitivity $\max_{i \in [n]} \|\mathbf{Ce}_i\|_2$ to the examples used in steps $j, b + j, \ldots, (k-1)b + j$ respectively. This is exactly Alg. 7 with the specified choice of $\Delta, \mathcal{S}$. $\qquad\square$

### F.3  Corollaries of Thm. 5

We give here several corollaries of Thm. 5 that are of interest.

**Equivalence to DP-SGD:**  Note that when $b = 1$, the partition contains a single subset, i.e., is the the entire dataset. In particular, in this setting Thm. 5 recovers the privacy guarantees of amplified DP-SGD under any amplification scheme, e.g. including the ones discussed below.

**Amplification via sampling:**  To recover Thm. 4 from Thm. 5, we take the distribution over $2^{[\tilde{m}]}$ corresponding to the uniform distribution over subsets of size $B$, and let $\mathcal{S}$ be the product of this distribution with itself $k$ times. This is equivalent to the following: in step $i$, we include each element of $D_{i \pmod{b}}$ independently with probability $q$. For this choice of $\mathcal{S}$, Alg. 6 reduces to Alg. 2 and Thm. 4. We next make the amplified privacy guarantee explicit in terms of the `dp_accounting` Python library [18]. Given $n, m, b$ and a target per-step batch size $B$, we could write a `dp_accounting.DpEvent` capturing the privacy guarantees of the matrix factorization mechanism as follows:

**Example F.1.**
```
gaussian_event = dp_accounting.GaussianDpEvent(noise_multiplier)
q = B / math.floor(n / b)
sampled_event = dp_accounting.PoissonSampledDpEvent(
    q, gaussian_event
)
composed_event = dp_accounting.SelfComposedDpEvent(
    sampled_event, math.ceil(m / b)
)
```

**Example F.2.** *To give an example of the amplification guarantee, for simplicity assume $n/b, m/b$ are integer. If all column norms in $\mathbf{C}$ are 1, each row of $\mathbf{x}$ has sensitivity 1, and each entry of $\mathbf{z}$ has standard deviation $\sigma$, then outputting $\mathbf{Cx} + \mathbf{z}$ satisfies $(\alpha, \frac{\alpha n}{2\sigma^2 b})$-RDP.*

*Using Theorem 11 of [46] and Thm. 4, for appropriate choice of $\alpha$ and $q$, this improves to $(\alpha, q^2 \cdot \frac{2\alpha n}{\sigma^2 b})$-RDP with amplification by sampling. In particular, if we have a target per-step batch size $B$, then we should choose $q = \frac{Bb}{m}$, and if this choice of $q$ satisfies the conditions in [46] plugging this in gives $(\alpha, \frac{2\alpha B^2 bn}{\sigma^2 m^2})$-RDP. Notice that $b = 1$ recovers the privacy guarantees of DP-SGD with sampling probability $B/m$, and this privacy guarantee weakens as $b$ increases.*

**Amplification via shuffling:**  Fix a per-step batch size $B$. Then, suppose we shuffle the list of examples, and cyclically iterate over batches of size $B$ in this list as the sets of examples to use in each step of matrix factorization. That is, we shuffle $D$ into an ordered list $d_1, d_2, \ldots$, and in step $i$ use examples $d_{(i-1)B+1 \pmod{m}}, d_{(i-1)B+2 \pmod{m}}, \ldots, d_{iB \pmod{m}}$.

For simplicity let's consider the case where $m/(Bb)$ is integer. In particular, this means in this shuffling scheme, each example appears once every $m/B$ steps, and for each of these steps $i$, $i \pmod{b}$ is the same. Then this shuffling scheme is equivalent to the following: First, rather than choose an arbitrary partition to apply Thm. 5, we choose a uniformly random partition into $b$ subsets

of size $m/b$. Then, we choose $\mathcal{S}$ to be the distribution giving by shuffling $[m/b]$ and then cyclically iterating over the shuffled list in batches of size $B$. Given this equivalence, we get the following:

**Corollary F.1.** *Suppose the examples in matrix factorization are chosen by shuffling $D$ and then iterating over batches of size $B$. If $n/(Bb)$ is integer, then the matrix factorization mechanism satisfies any standard privacy guarantee satisfied by $k$ adaptive scalar queries with sensitivity $\max_{i\in[n]} \|\mathbf{C}\mathbf{e}_i\|_2$ and noise $N(0, \sigma^2)$, with the examples in each query given by shuffling a dataset of size $m/b$ and cyclically iterating over this list in batches of size $B$.*

**Example F.3.** *Consider the simplified case where $m = n$, we choose a random permutation $\pi$, and in step $i$ query example $d_{\pi(i)}$. In this case, if all the column norms of $\mathbf{C}$ are 1, $\mathbf{x}$'s rows have sensitivity 1, and $\mathbf{z}$'s entries have standard deviation $\sigma = \mathcal{O}\left(\frac{\sqrt{\ln(1/\delta)}}{\epsilon}\right)$, we get that $\mathbf{C}\mathbf{x} + \mathbf{z}$ satisfies $(\epsilon, \delta)$-DP. With e.g., the amplification for shuffled $(\epsilon, \delta)$-DP mechanisms given by Theorem 5.1 of [6] and Cor. F.1, if $\epsilon$ is a constant, we instead get that $\mathbf{C}\mathbf{x} + \mathbf{z}$ satisfies $\left(\epsilon \cdot \mathcal{O}\left(\sqrt{\frac{b \log(1/\delta)}{n}}\right), \delta \cdot \mathcal{O}\left(\frac{n \ln(1/\delta)}{b}\right)\right)$-DP.*

### F.4 Optimizing the number of bands

Let $\sigma_{\epsilon,\delta}(b)$ be the required Gaussian noise magnitude for a $b$-banded MF run for $n$ iterations using e.g. Alg. 2 to achieve $(\epsilon, \delta)$-DP with per-step batch size $B$. Then, the expected total squared error introduced while achieving $(\epsilon, \delta)$-DP with amplification can be calculated as

$$\sigma_{\epsilon,\delta}(b)^2 \mathcal{L}(\mathbf{A}\mathbf{C}_b^{-1}, \mathbf{C}_b)$$

where $\mathbf{C}_b$ is a $b$-banded lower triangular matrix optimized via Problem 2. Generally, smaller values of $b$ will allow for more amplification, and hence a smaller $\sigma$; however, this introduces a stronger set of constraints on the optimization problem, likely increasing the $\mathcal{L}$ term. Hence, the choice of $b$ should be optimized. Fortunately, $\sigma_{\epsilon,\delta}(\cdot)$ can be computed efficiently: Thm. 4 implies a procedure to compute $\epsilon$ given $\sigma, \delta, b$, and then one can use binary search[10] and this procedure to find the $\sigma$ giving a desired $\epsilon$. In addition, one can pre-compute the optimal matrices $\mathbf{C}_b$ for different numbers of bands. The search can be restricted to $b \in \{1, 2, \ldots, \frac{m}{B}, n\}$ since for $b = \frac{m}{B}$ we have $|D_j| = B$, i.e. Alg. 2 and Thm. 4 provide no privacy amplification.

Unlike the un-amplified version of MF, now the best factorization depends on the privacy parameters $(\epsilon, \delta)$. The benefits of amplification are generally stronger for small $\epsilon$: For example, amplification by sampling with probability $p$ roughly improves $\epsilon$ to $\log(1 + p(e^\epsilon - 1))$ (see e.g. Section 6 of [55]), which is approximately $p\epsilon$ for $\epsilon \leq 1$, and approximately $\epsilon - \log(1/p)$ if $e^\epsilon \gg 1/p$. Hence, with smaller values of $\epsilon$, we expect the benefits of amplification to outweigh the benefits of correlated noise, in which case $b = 1$ will be optimal. With larger values of $\epsilon$, we expect the benefits of correlated noise to outweigh the benefits of amplification, and in this regime $b = n$ will be optimal. For moderate values of $\epsilon$, we expect the optimal $b$ to be somewhere in the middle.

### F.5 Applying BANDMF with Privacy Amplification

Consider the setup of Sec. 5 with our CIFAR10 setting described fully in App. H. As mentioned in that section, we use the convention that our privacy analysis will assume Poisson sampling, even though we are using passes over a shuffled dataset. We have $m = 50,000$ and train for $k = 20$ epochs with a batch size $B = 500$. Choquette-Choo et al. [15] lets us bound the sensitivity and optimize matrices for this setting, however, without privacy amplification. Suppose we choose $\hat{b} = 100$. Because $m/\hat{b} = 500 = B$, we get that the sampling probability is 100% (see Example F.1) and thus get no benefits from amplification. For all $\hat{b} \in [1, 100)$ we get amplification benefits which can be seen intuitively as follows.

If $\hat{b} = 2$, we get that there are two partitions $D_1, D_2$. Then or first event will be the simultaneous release of $\mathbf{x}_1, \mathbf{x}_2$, our second of $\mathbf{x}_3, \mathbf{x}_4$, and so on. Because each partition is of size $|D_j| = m/\hat{b} = 25,000$ and $B = 500$, we have a sampling probability $q = 2\%$. Given our parameters, we also have $d = k \cdot m/B = 2,000$, and so we must compose $d/\hat{b} = 1,000$ events (as seen in Example F.1). Because in this setting each event is the batch release of $\hat{b}$ steps of $\mathbf{x}$, where each example participates at most

---

[10]See for example the `calibrate_dp_mechanism` function of DP Team [18].

once on each release, observe that we need only normalize the sensitivity of this mechanism under $(k = 1, b = \hat{b} = 2)$-participation. Generally, as $\hat{b}$ increases we have a higher sampling probability, but fewer events to compose, and vice versa. As can be seen, when $\hat{b} = 1$, this reduces to the standard accounting for DP-SGD. It can also be seen that we desire each $D_j$ to be a non-overlapping partition of $D$, as otherwise, a single example may participate multiple times in the same event (and thus have a higher sampling probability).

# G    Additional RMSE Experiment Details

## G.1    Optimal Number of Bands

In this section, we provide supplementary data surrounding the RMSE experiments in Fig. 5. Table 3 shows the optimal number of bands for each $(\epsilon, k)$ pair considered in the RMSE experiments. It shows the general trend that as $\epsilon$ decreases, or $k$ increases, the optimal number of bands decreases.

## G.2    Explaining many-epoch setting

Observe in Fig. 5 that as BANDMF and MULTI-EPOCH MF incur similar RMSE as DP-SGD when the number of epochs increases.

This phenomenon occurs because as the number of epochs changes, so does the sensitivity of the $\mathbf{X}$ matrix, and subsequently the geometry of the optimization problem. By Eq. (5), we see that sensitivity can be calculated as a sum of absolute values of entries of $\mathbf{X}$ corresponding to iterations where a single user might participate. As the number of epochs increases, the number of entries of $\mathbf{X}$ that we have to sum up also increases. It turns out that under the constraint of constant sensitivity, it is better to put more weight on the diagonal of $\mathbf{X}$ than the off-diagonal. This is an observation we have made by solving this optimization problem numerically in a number of settings. Hence, as the number of epochs increases, $\mathbf{X}$ becomes more and more diagonally dominant, making the mechanism closer to DP-SGD ($\mathbf{X}$ = Identity).

| $\epsilon/k$ | 1 | 2 | 4 | 8 | 16 | 32 | 64 | 128 | 256 | 512 | 1024 |
|---|---|---|---|---|---|---|---|---|---|---|---|
| 0.03125 | 2 | 2 | 1 | 1 | 1 | 1 | 1 | 1 | 1 | 1 | 1 |
| 0.0625 | 4 | 2 | 1 | 1 | 1 | 1 | 1 | 1 | 1 | 1 | 1 |
| 0.125 | 8 | 4 | 2 | 1 | 1 | 1 | 1 | 1 | 1 | 1 | 1 |
| 0.25 | 8 | 4 | 4 | 2 | 1 | 1 | 1 | 1 | 1 | 1 | 1 |
| 0.5 | 16 | 8 | 4 | 4 | 2 | 1 | 1 | 1 | 1 | 1 | 1 |
| 1.0 | 32 | 16 | 8 | 4 | 2 | 2 | 1 | 1 | 1 | 1 | 1 |
| 2.0 | 64 | 32 | 16 | 8 | 4 | 2 | 2 | 1 | 1 | 1 | 1 |
| 4.0 | 128 | 64 | 32 | 16 | 8 | 4 | 2 | 2 | 1 | 1 | 1 |
| 8.0 | 1024 | 512 | 256 | 32 | 16 | 8 | 4 | 2 | 2 | 1 | 1 |
| 16.0 | 1024 | 512 | 256 | 128 | 64 | 32 | 8 | 4 | 4 | 2 | 1 |

Table 3:  Optimal number of bands for each $(\epsilon, k)$ pair, when $n = 1024$ and $\delta = 10^{-6}$.

# H    Additional CIFAR-10 Experiment Details

## H.1    Setup and Tuning

We tune all jobs on a learning rate grid of coefficients in $\{1, 2, 5\}$ on powers in [-2, 3]. We find that no momentum works best for DP-SGD and momentum=0.95 works best for MF-DP-FTRL mechanisms on average in tuning; though initial tuning found that tuning momentum as well could lead to slightly better results at some $\epsilon$ budgets, we found that a more refined grid of learning rates nearly always led to a fixed momentum being optimal, and so we fix this parameter. We also found that a learning rate cooldown to $0.05\times$ the initial learning rate over the last 500 steps of training improved all runs and so we fix this parameter. All models trained for 20 epochs on CIFAR10 with a batch size of 500. We repeat each setting 12 times and show 95% bootstrapped confidence intervals.

## H.2 Additional Figures

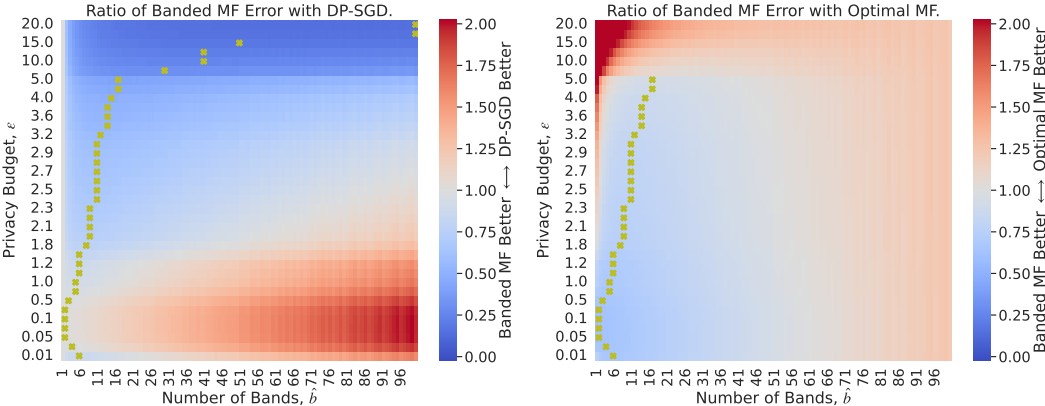

Figure 11: **On CIFAR-10, BANDMF is at least as good as DP-SGD across all $\epsilon$, and often signifi-cantly better.** BANDMF is better than the prior MF-DP-FTRL from Choquette-Choo et al. [15] up to $\epsilon \approx 5$. We compare the ratio of the total error (see Sec. 4) of BANDMF with either mechanism. Lower values indicate that BANDMF is better. The yellow markers indicate the best BANDMF mechanism that was better for that $\epsilon$ budget if one existed. Unlike in Fig. 5(b), We only optimize the Band MF over $\hat{b} \in [0, n/k]$ which leads to a regime around $\epsilon > 5$ where the it performs worse than the Multi-epoch MF of Choquette-Choo et al. [15].

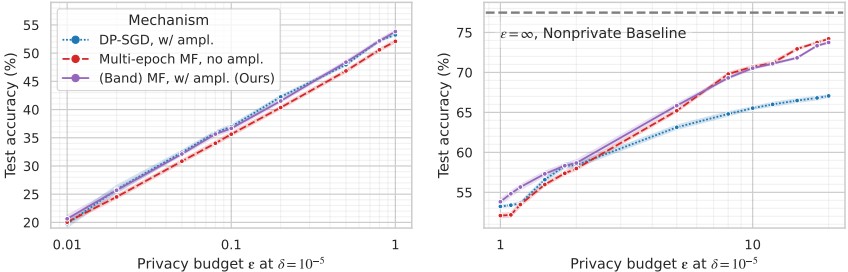

Figure 12: **Our banded matrices consistently perform at least as well as the best prior method in each range of $\epsilon$.** Around $\epsilon \approx 1$, we observe significant utility benefits from the banded mechanism around $2-3$ percentage points over DP-SGD. We only optimize the Band MF over $\hat{b} \in [0, n/k]$ which leads to a regime around $\epsilon > 5$ where the it performs worse than the Multi-epoch MF of Choquette-Choo et al. [15]; $\hat{b} = n$ is equivalent to this approach modulo the sensitivity definition which we exclude to emphasize the regime we improve on. Empirical setup is in App. H.

# I  Additional StackOverflow Next-Word-Prediction Experiment Details

We follow the experimental setup for StackOverflow NWP from Denisov et al. [17] and Choquette-Choo et al. [15]. Except for SINGLE-EPOCH MF (which uses $B = 167$ clients/round for 1 epoch), all privacy guarantees and accuracy results are for 6 epochs of training using $B = 1000$ clients/round for 2052 rounds (also 1 epoch). The matrices used in these experiments are included in Table 2.

For computational efficiency in estimating model accuracy at a given privacy guarantee, we actually compute in simulation updates from only 100 clients/round, and scale the noise multiplier by a corresponding factor ($\frac{100}{1000}$ for 6 epoch experiments, $\frac{100}{167}$ for SINGLE-EPOCH MF). This approach has been used previously [35, 44], and we independently verified it has a negligible impact on the estimates of accuracy figures we report. Tables 4 and 5 include the unscaled noise multipliers $\sigma$ for our experiments.

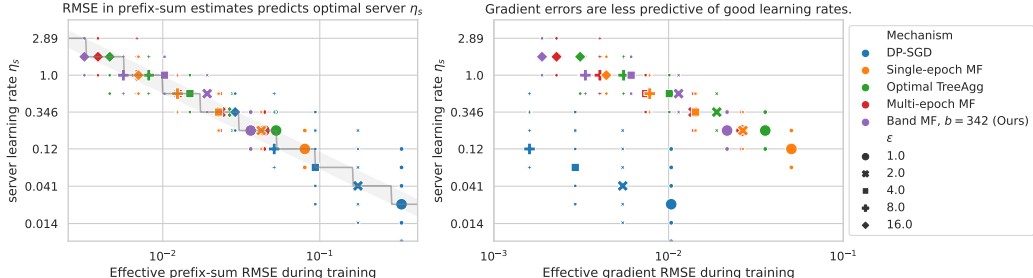

Figure 13: Correlation between optimal server learning rates $\eta_s$ and the effective RMSE during training, see Eq. (11).

**Optimizer and learning-rate tuning** For all SO NWP experiments we use the FedSGDM optimizer [50]. This optimization approach first takes multiple local SGD steps (with learning rate 1.0 in our experiments) on the training data of each user in the batch (cohort) before clipping to $\zeta = 1$, summing, and passing the result $\mathbf{x}_i$ into the DP mechanism which adds noise $[\mathbf{C}^\dagger \mathbf{z}]_{[i,:]} \in \mathbb{R}^d$ on each iteration $i$. The resulting privatized sum is then divided by the batch size $B$ and passed to the "server" (post-aggregation) optimizer, in our case SGDM with momentum parameter $\beta = 0.95$ and learning rate $\eta_s$. We find tuning $\eta_s$ depending on the noise level is critical. By using the computationally efficient approach mentioned above, we were able to conduct rigorous tuning over a learning rate grid of $1.7^i$ for powers $i$ in $\{-9, \ldots, 4\}$, estimating good initial guesses based on prior work. Table 6 gives the full set of results, and Fig. 14 shows convergence as a function of the number of rounds (iters).

**Learning rate warmup and cooldown** Denisov et al. [17] found learning rate cooldown was effective, and Choquette-Choo et al. [15] found that zeroing-out client updates with large $\ell_\infty$ norms was critical to stability in early training. We find that additionally introducing a learning-rate warmup schedule reduces the need for this zeroing-out (though we still enable it), and generally decreases the variance in training results. Hence, all of our experiments (for all algorithms) using a linear learning rate warmup from $0.05\eta_s$ to $1.0\eta_s$ over the first $15\%$ of rounds (309), and a linear decay from $1.0\eta_s$ to $0.05\eta_s$ over the last $25\%$ of rounds (513).

**Using RMSE to tune optimal server learning rates** Fig. 13 plots the server learning rates $\eta_s$ from Table 6 on the $y$-axis (with the optimal rates shown as larger symbols, and sub-optimal rates as small symbols, versus two different measures of the error for the DP mechanism on the $x$-axis: The left plot gives uses the effective prefix-sum RMSE (the objective we use for optimizing (banded) matrices $\mathbf{C}$),

$$\text{(Mechanism error)} \times \text{noise-multiplier/(clients-per-round)} = \sqrt{\mathcal{L}(\mathbf{SC}^{-1}, \mathbf{C})/n} \times \sigma/B, \quad (11)$$

where $\mathbf{S}$ is the prefix-sum workload (lower-triangular matrix of ones) and $\sigma$ and $B$ are as given in Table 4. The right plot uses the RMSE in error of individual gradients, computed by replacing the $\mathcal{L}$ term in the above with $\mathcal{L}(\mathbf{IC}^{-1}, \mathbf{C})$ where we take the workload $\mathbf{A}$ to be the identity matrix $\mathbf{I}$ rather than the prefix sum matrix $\mathbf{S}$.

We see a strong linear correlation between the prefix-sum RMSE and optimal learning rate in the left plot; this does not hold for individual gradient errors (right plot). Based on this, we use the following linear regression to choose learning rates for the non-federated (amplified) SO NWP experiments (still rounding to the nearest $1.7^i$ for consistency):

$$\log(\eta_s) = -0.95 \cdot \log(L_e) - 4.64$$

This allowed us to estimate learning rates for the amplified experiments with a high degree of accuracy; Table 7 gives the final selected learning rates.

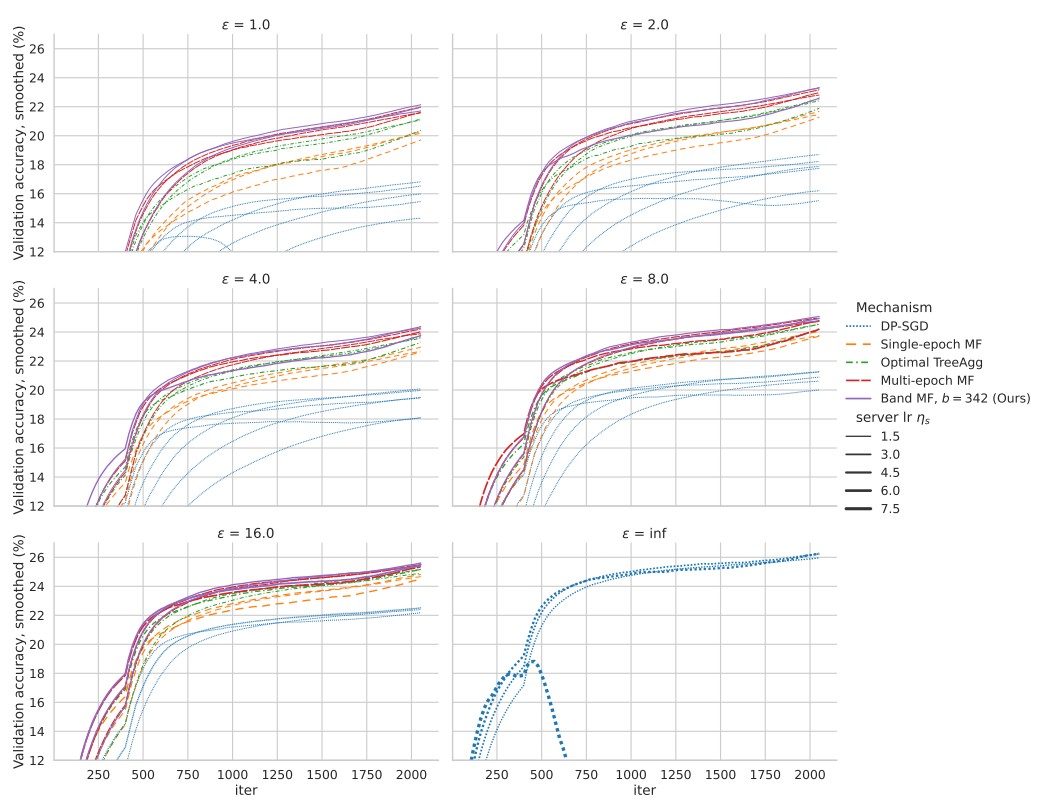

Figure 14: Convergence plots for all cross-device federated learning simulation experiments.

| Mechanism | clients per round $B$ | $\epsilon$ | noise mult. $\sigma$ | server lr $\eta_s$ | Eval Accuracy (%, Smoothed) | Test Accuracy (%) |
|---|---|---|---|---|---|---|
| DP-SGD | 1000 | 1 | 4.22468 | 0.0244 | 16.82 | 16.69 |
| Single-epoch MF | 167 | 1 | 4.22468 | 0.1197 | 20.29 | 20.44 |
| Optimal TreeAgg | 1000 | 1 | 4.22468 | 0.2035 | 21.15 | 21.25 |
| Multi-epoch MF | 1000 | 1 | 4.76079 | 0.2035 | 21.96 | 21.92 |
| Band MF (Ours) | 1000 | 1 | 4.22468 | 0.2035 | 22.12 | 22.05 |
| DP-SGD | 1000 | 2 | 2.23048 | 0.0414 | 18.70 | 18.42 |
| Single-epoch MF | 167 | 2 | 2.23048 | 0.2035 | 21.66 | 21.70 |
| Optimal TreeAgg | 1000 | 2 | 2.23048 | 0.3460 | 22.52 | 22.59 |
| Multi-epoch MF | 1000 | 2 | 2.51352 | 0.3460 | 23.15 | 23.04 |
| Band MF (Ours) | 1000 | 2 | 2.23048 | 0.5882 | 23.31 | 23.19 |
| DP-SGD | 1000 | 4 | 1.19352 | 0.0704 | 20.07 | 19.81 |
| Single-epoch MF | 167 | 4 | 1.19352 | 0.3460 | 22.94 | 22.90 |
| Optimal TreeAgg | 1000 | 4 | 1.19352 | 0.5882 | 23.66 | 23.62 |
| Multi-epoch MF | 1000 | 4 | 1.34498 | 0.5882 | 24.19 | 24.02 |
| Band MF (Ours) | 1000 | 4 | 1.19352 | 1.0000 | 24.35 | 24.16 |
| DP-SGD | 1000 | 8 | 0.65294 | 0.1197 | 21.26 | 21.08 |
| Single-epoch MF | 167 | 8 | 0.65293 | 0.5882 | 24.03 | 23.88 |
| Optimal TreeAgg | 1000 | 8 | 0.65294 | 1.0000 | 24.54 | 24.45 |
| Multi-epoch MF | 1000 | 8 | 0.73579 | 1.0000 | 24.95 | - |
| Band MF (Ours) | 1000 | 8 | 0.65294 | 1.0000 | 25.06 | 24.88 |
| DP-SGD | 1000 | 16 | 0.36861 | 0.3460 | 22.51 | 22.26 |
| Single-epoch MF | 167 | 16 | 0.36861 | 1.0000 | 24.80 | 24.62 |
| Optimal TreeAgg | 1000 | 16 | 0.36861 | 1.7000 | 25.15 | 25.14 |
| Multi-epoch MF | 1000 | 16 | 0.41539 | 1.7000 | 25.50 | 25.33 |
| Band MF (Ours) | 1000 | 16 | 0.36861 | 1.7000 | 25.59 | 25.41 |

Table 4: Parameters and metrics for **simulated cross-device FL**, Fig. 6[a]. The noise multipliers $\sigma$ are calibrated to achieve the given $\epsilon$ guarantees at $\delta=10^{-6}$ under $b=342$-min-separation. The matrices are scaled to have sensitivity 1 under $(k=6, b=342)$, see Table 2, and so a larger noise multiplier $\sigma$ is necessary for the MULTI-EPOCH MF matrices. Test-set accuracy for MULTI-EPOCH MF at $\epsilon = 8$ was unavailable. All BANDMF matrices use $\hat{b}=342$. Note that $\sigma$ is reported for reproducibility purposes (it is the parameter passed to Alg. 1), but it does *not* directly capture the noise added in either gradients $\hat{x}_i$ or their cumulative sums as the noise $z$ is linearly transformed by $C^{-1}$ which varies across the mechanisms above.

| Mechanism | clients per round $B$ | $\epsilon$ | noise mult. $\sigma$ | server lr $\eta_s$ | Eval Accuracy (%, Smoothed) | Test Accuracy (%) |
|---|---|---|---|---|---|---|
| DP-SGD, w/ ampl. | 1000 | 1 | 0.37313 | 0.3460 | 22.50 | 22.22 |
| Multi-epoch MF, no ampl. | 1000 | 1 | 4.22468 | 0.2035 | 22.11 | 22.10 |
| (Band) MF, w/ ampl. (Ours) | 1000 | 1 | 0.79118 | 0.3460 | 23.11 | 22.83 |
| DP-SGD, w/ ampl. | 1000 | 2 | 0.30481 | 0.3460 | 22.89 | 22.62 |
| Multi-epoch MF, no ampl. | 1000 | 2 | 2.23048 | 0.3460 | 23.36 | 23.24 |
| (Band) MF, w/ ampl. (Ours) | 1000 | 2 | 0.64708 | 0.5882 | 24.01 | 23.71 |
| DP-SGD, w/ ampl. | 1000 | 4 | 0.25136 | 0.3460 | 23.27 | 22.94 |
| Multi-epoch MF, no ampl. | 1000 | 4 | 1.19352 | 0.5882 | 24.36 | 24.16 |
| (Band) MF, w/ ampl. (Ours) | 1000 | 4 | 0.52224 | 1.0000 | 24.67 | 24.42 |
| DP-SGD, w/ ampl. | 1000 | 8 | 0.20567 | 0.5882 | 23.59 | 23.30 |
| Multi-epoch MF, no ampl. | 1000 | 8 | 0.65294 | 1.0000 | 25.08 | 24.88 |
| (Band) MF, w/ ampl. (Ours) | 1000 | 8 | 0.43490 | 1.7000 | 25.26 | 24.99 |
| DP-SGD, w/ ampl. | 1000 | 16 | 0.16876 | 0.5882 | 23.96 | 23.61 |
| Multi-epoch MF, no ampl. | 1000 | 16 | 0.36861 | 1.7000 | 25.59 | 25.43 |
| (Band) MF, w/ ampl. (Ours) | 1000 | 16 | 0.36861 | 1.7000 | 25.59 | 25.43 |

Table 5: Parameters and metrics for **centralized StackOverflow**, Fig. 1(b). The noise multipliers are calibrated to achieve the given $\epsilon$ guarantees at $\delta{=}10^{-6}$ under $(k{=}6, b{=}342)$-participation, assuming Poisson sampling for DP-SGD and BANDMF. For BANDMF, we tune $\hat{b}$ under amplification for optimal RMSE, selecting $\hat{b} = 9, 18, 32, 64, 2052$ for $\epsilon = 1, 2, 4, 8, 16$ respectively. For $\epsilon = 16$, we have $n = \hat{b}$, and hence BANDMF is identical to MULTI-EPOCH MF optimized with Eq. (6). Values of $\sigma$ are provided for reproducibility only, see comments on $\sigma$ from Table 4.

| ε | Mechanism | Eval Accuracy (%, Smoothed) | | | | | | | | | | | | | |
|---|---|---|---|---|---|---|---|---|---|---|---|---|---|---|---|
| | server lr $\eta_s$ | 0.0084 | 0.0143 | 0.0244 | 0.0414 | 0.0704 | 0.1197 | 0.2035 | 0.3460 | 0.5882 | 1.0000 | 1.7000 | 2.8900 | 4.9130 | 8.3521 |
| 1.0 | DP-SGD | 14.31 | 15.98 | **16.82** | 16.53 | 15.46 | 4.67 | - | - | - | - | - | - | - | - |
| | Single-epoch MF | - | - | - | - | 20.16 | **20.29** | 19.68 | - | - | - | - | - | - | - |
| | Optimal TreeAgg | - | - | - | - | - | 21.08 | **21.15** | 20.34 | - | - | - | - | - | - |
| | Multi-epoch MF | - | - | - | - | - | 21.56 | **21.96** | 21.60 | - | - | - | - | - | - |
| | Band MF, $b$=342 (Ours) | - | - | - | - | - | 21.70 | **22.12** | 21.96 | - | - | - | - | - | - |
| 2.0 | DP-SGD | - | 16.20 | 17.88 | **18.70** | 18.22 | 17.75 | 15.52 | - | - | - | - | - | - | - |
| | Single-epoch MF | - | - | - | - | - | 21.46 | **21.66** | 21.26 | - | - | - | - | - | - |
| | Optimal TreeAgg | - | - | - | - | - | - | 22.40 | **22.52** | 21.87 | - | - | - | - | - |
| | Multi-epoch MF | - | - | - | - | - | - | 22.80 | **23.15** | 22.96 | - | - | - | - | - |
| | Band MF, $b$=342 (Ours) | - | - | - | - | - | - | - | 23.27 | **23.31** | 22.57 | - | - | - | - |
| 4.0 | DP-SGD | - | - | 18.08 | 19.45 | **20.07** | 19.97 | 19.48 | 18.08 | - | - | - | - | - | - |
| | Single-epoch MF | - | - | - | - | - | - | 22.66 | **22.94** | 22.60 | - | - | - | - | - |
| | Optimal TreeAgg | - | - | - | - | - | - | - | 23.57 | **23.66** | 23.27 | - | - | - | - |
| | Multi-epoch MF | - | - | - | - | - | - | - | 23.87 | **24.19** | 24.01 | - | - | - | - |
| | Band MF, $b$=342 (Ours) | - | - | - | - | - | - | - | - | 24.26 | **24.35** | 23.74 | - | - | - |
| 8.0 | DP-SGD | - | - | - | - | 20.61 | **21.26** | 21.24 | 20.89 | 20.00 | - | - | - | - | - |
| | Single-epoch MF | - | - | - | - | - | - | - | 23.73 | **24.03** | 23.71 | - | - | - | - |
| | Optimal TreeAgg | - | - | - | - | - | - | - | - | 24.52 | **24.54** | 24.15 | - | - | - |
| | Multi-epoch MF | - | - | - | - | - | - | - | - | 24.72 | **24.95** | 24.77 | 24.17 | - | - |
| | Band MF, $b$=342 (Ours) | - | - | - | - | - | - | - | - | 24.76 | **25.06** | 24.92 | - | - | - |
| 16.0 | DP-SGD | - | - | - | - | - | - | 22.39 | **22.51** | 22.17 | - | - | - | - | - |
| | Single-epoch MF | - | - | - | - | - | - | - | - | 24.66 | **24.80** | 24.50 | - | - | - |
| | Optimal TreeAgg | - | - | - | - | - | - | - | - | 24.89 | 25.15 | **25.15** | - | - | - |
| | Multi-epoch MF | - | - | - | - | - | - | - | - | - | 25.38 | **25.50** | 25.34 | - | - |
| | Band MF, $b$=342 (Ours) | - | - | - | - | - | - | - | - | - | 25.38 | **25.59** | 25.47 | - | - |
| inf | DP-SGD | - | - | - | - | - | - | - | - | - | - | 25.96 | 26.23 | **26.24** | 8.03 |

Table 6: **Federated learning rate tuning for StackOverflow NWP.** Validation accuracy smoothed over the final 400 rounds of training, used to select the best server learning rates for the comparison of test-set accuracy presented in Fig. 6[a].

| $\epsilon$ | server lr $\eta_s$ / Mechanism | Eval Accuracy (%, Smoothed) | | | | | | | |
| --- | --- | --- | --- | --- | --- | --- | --- | --- | --- |
| | | 0.1197 | 0.2035 | 0.3460 | 0.5882 | 1.0000 | 1.7000 | 2.8900 | 4.9130 |
| 1.0 | **DP-SGD** | - | 22.39 | **22.50** | 22.03 | - | - | - | - |
| | **Multi-epoch MF** | 21.75 | **22.11** | 21.95 | - | - | - | - | - |
| | **Band MF, $b$=9 (Ours)** | - | 22.83 | **23.11** | 23.03 | - | - | - | - |
| 2.0 | **DP-SGD** | - | 22.70 | **22.89** | 22.66 | - | - | - | - |
| | **Multi-epoch MF** | - | 22.89 | **23.36** | 23.26 | - | - | - | - |
| | **Band MF, $b$=18 (Ours)** | - | - | 23.80 | **24.01** | 23.77 | - | - | - |
| 4.0 | **DP-SGD** | - | 22.88 | **23.27** | 23.20 | - | - | - | - |
| | **Multi-epoch MF** | - | - | 23.96 | **24.36** | 24.22 | 23.71 | - | - |
| | **Band MF, $b$=32 (Ours)** | - | - | - | 24.52 | **24.67** | 24.43 | - | - |
| 8.0 | **DP-SGD** | - | - | 23.48 | **23.59** | 23.28 | - | - | - |
| | **Multi-epoch MF** | - | - | - | 24.79 | **25.08** | 24.98 | 24.55 | - |
| | **Band MF, $b$=64 (Ours)** | - | - | - | - | 25.15 | **25.26** | 24.79 | - |
| 16.0 | **DP-SGD** | - | - | 23.85 | **23.96** | 23.72 | - | - | - |
| | **Multi-epoch MF** | - | - | - | - | 25.42 | **25.59** | 25.50 | 24.92 |
| | **Band MF, $b$=342 (Ours)** | - | - | - | - | 25.37 | **25.55** | 25.45 | 24.90 |
| | **Band MF, $b$=64 (Ours)** | - | - | - | - | 25.38 | **25.54** | 25.40 | - |

Table 7: **Centralized learning rate tuning for StackOverflow NWP..** Validation accuracy smoothed over the final 400 rounds of training, used to select the best server learning rates for the comparison of test-set accuracy presented in Fig. 1(b). DP-SGD and BANDMF use amplification.

| **Algorithm 8** Banded Matrix Multiplication | **Algorithm 9** Banded Inverse Multiplication |
|---|---|
| **Input:** $\hat{b}$-Banded lower triangular matrix $\mathbf{C} \in \mathbb{R}^{n \times n}$, vector $\mathbf{x} \in \mathbb{R}^n$ 
 **Output:** $\mathbf{Cx}$ 
 **for** $i = 1, \ldots, n$ **do** 
 $\quad \mathbf{y}_i = \sum_{j=i-\hat{b}+1}^{i} \mathbf{C}_{[i,j]} \mathbf{x}_j$ 
 **return y** | **Input:** $\hat{b}$-Banded lower triangular matrix $\mathbf{C} \in \mathbb{R}^{n \times n}$, vector $\mathbf{y} \in \mathbb{R}^n$ 
 **Output:** $\mathbf{C}^{-1}\mathbf{y}$ 
 **for** $i = 1, \ldots, n$ **do** 
 $\quad \mathbf{x}_i = (\mathbf{y}_i - \sum_{j=i-\hat{b}+1}^{i-1} \mathbf{C}_{[i,j]} \mathbf{x}_j)/\mathbf{C}_{[i,i]}$ 
 **return x** |

Figure 15: Algorithms for matrix-vector and inverse matrix-vector multiplication by a banded matrix. To simplify the presentation, we use the convention that out-of-bounds indexing into a matrix or vector returns 0.

## J   Efficient Multiplication and Inverse of Banded Matrices

Algorithms 8 and 9 (Fig. 15) give algorithms for lower triangular banded matrix-vector multiplication and inverse banded matrix-vector multiplication. Note that both algorithms are compatible with the streaming nature of gradients. As soon as the next input $\mathbf{x}_i$ is received, the algorithm can immediately output $\mathbf{y}_i$. Both algorithms require storing a state of size $\hat{b}$, and run in $O(n \cdot \hat{b})$ time. While the algorithms are described with respect to computing matrix-vector products, they can also be used to compute matrix-matrix products where the right-hand-side is a $n \times d$ matrix by multiplying by each column independently. In this setting, these algorithms require $O(\hat{b} \cdot d)$ space and $O(n \cdot \hat{b} \cdot d)$ time. Both algorithms have appeared previously in the literature on Monte Carlo methods, which have a similar problem at their core to that of noise generation for MF; see e.g. [56, Section 2].

## K   Application to a Real-World Cross-Device FL System

We train a one-layer LSTM language model of ~2.4 million parameters in a practical cross-device FL system following [58] . The model is used for predicting the next word of Spanish in a mobile virtual keyboard. We pretrain the model on public multilingual C4 dataset [49, 60], and then fine-tune with on-device user data in FL. In a common practical FL system, clients have to satisfy criteria like being charged, idle and connected to unmetered network to participate in a round [10, 27, 31, 47], hence only a subset of clients can be reached and there is a strong diurnal pattern of client participation [61, 63]. It is very challenging to hold a fixed set of clients for evaluation, or develop random sampling for privacy amplification. Though the current implementation of client participation control is feasible through the client timer, the tuning of separation $b$ in practice can be challenging. Therefore, the training of Fig. 6 (b) can achieve smaller separation $b$ than in [58].

### K.1   Reporting privacy guarantees

We follow the guidelines outlined in [48, Sec. 5.3] to report privacy guarantees.

1. **DP setting**. This a central DP guarantee where the service provider is trusted to correctly implement the mechanism.
2. **Instantiating the DP Definition**
   (a) *Data accesses covered*: The DP guarantee applies to all well-behaved clients [11] in a single training run. We do not account for hyperparameter tuning, or the selection of the final model checkpoint using evaluation metrics or A/B testing in our guarantees. Public multilingual C4 data [49, 60] is used for pre-training.
   (b) *Final mechanism output*: Only the final model checkpoint is released for use in production, however the mechanism's output is technically the full sequence of privatized gradients, and so the guarantee also applies at this level, and hence all intermediate models are protected (including those sent to devices participating in federated learning).

---

[11]Clients that faithfully follow the algorithm including participation limits. Due to the design of the algorithm, a mis-behaved client does not adversely affect the DP guarantee of any well-behaved clients.

(c) *Unit of privacy.* Device-level DP is considered, i.e., the notion of adjacency is with respect to arbitrary training datasets on each client device, and the device might have an arbitrarily large local dataset containing arbitrary training examples. For user's with a single device, this corresponds directly to user-level DP; for devices shared with multiple users, this provides a stronger notion of DP than user-level; for a user with multiple devices that happen to both participate in training the model, the notion is weaker, but group privacy can be used to obtain a user-level guarantee.

(d) *Adjacency definition for "neigbouring" datasets*: We use the zero-out definition [35]. This is a a special form of the add-or-remove definition, where neighboring data sets differ by addition/removal of a single client. In the absence of a client at any training step, we assume that the client's model update gets replaced with the all zeros vector. This assumption enforces a subtle modification to the traditional definition of the add/remove notion of DP which allows neighboring data sets to have the same number of records.

3. **Privacy accounting details**

(a) *Type of accounting used*: Both $\rho-$zCDP [11] accounting, and PLD accounting [18] for $(\epsilon, \delta)-$DP are used.

(b) *Accounting assumptions* : Each client only participates limited times during the training, and there are at least a min-separation of $b$ rounds between two consecutive participation of a client. This is enforced by a timer on clients in the cross-device FL system.

(c) *The formal DP statement*: The privacy guarantees are $\rho$=0.52-zCDP and $(\epsilon$=6.69, $\delta$=$10^{-10})$-DP for ONLINE TREEAGG, while BANDMF achieves $\rho$=0.24-zCDP and $(\epsilon$=4.35, $\delta$=$10^{-10})$-DP.

(d) *Transparency and verifiability*: We are going to open source our code based on Tensor-Flow Federated and Tensorflow Privacy. Key portions of the cross-device FL system will also open sourced.

