:** Take the distribution over $2^{[\tilde{m}]}$ given by including each element of $[\tilde{m}]$ independently with probability $q$, and let $\mathcal{S}$ be the product of this distribution with itself $k$ times. This is equivalent to the following: in round $i$, we include each element of $D_{i \pmod b}$ independently with probability $q$. In particular, within each $D_j$, we are just using sampling with replacement to

choose which elements to include in each round. From this we get the following corollary, which allows us to reduce to a setting whose privacy guarantees are well-understood:

**Corollary E.1.** *Suppose the examples participating in round $i$ of matrix factorization are chosen by including each element of $D_{i \pmod b}$ independently with probability q. Then the matrix factorization mechanism satisfies any standard privacy guarantee satisfied by $k$ adaptive scalar queries with sensitivity $\max_{i \in [n]} \|\mathbf{C}\mathbf{e}_i\|_2$ and noise $N(0, \sigma^2)$, with the $i$th query run on a batch given by sampling each element of a $\breve{m}$-element database with probability q.*

We next make this explicit in terms of the `dp_accounting` Python library [18]. Given $n, m, b$ and a target per-round batch size $B$, we could write a `dp_accounting.DpEvent` capturing the privacy guarantees of the matrix factorization mechanism as follows:

```
gaussian_event = dp_accounting.GaussianDpEvent(noise_multiplier)
q = B / math.floor(n / b)
sampled_event = dp_accounting.PoissonSampledDpEvent(
    q, gaussian_event
)
composed_event = dp_accounting.SelfComposedDpEvent(
    sampled_event, math.ceil(m / b)
)
```

**Example E.1.** *To give an example of the amplification guarantee, for simplicity assume $n/b, m/b$ are integer. If all column norms in $\mathbf{C}$ are 1, each row of $\mathbf{x}$ has sensitivity 1, and each entry of $\mathbf{z}$ has standard deviation $\sigma$, then outputting $\mathbf{C}\mathbf{x} + \mathbf{z}$ satisfies $(\alpha, \frac{\alpha n}{2\sigma^2 b})$-RDP.*

*Using Theorem 11 of [43] and Cor. E.1, for appropriate choice of $\alpha$ and q, this improves to $(\alpha, q^2 \cdot \frac{2\alpha n}{\sigma^2 b})$-RDP with amplification by sampling. In particular, if we have a target per-round batch size $B$, then we should choose $q = \frac{Bb}{m}$, and if this choice of q satisfies the conditions in [43] plugging this in gives $(\alpha, \frac{2\alpha B^2 bn}{\sigma^2 m^2})$-RDP. Notice that $b = 1$ recovers the privacy guarantees of DP-SGD with Poisson sampling, and this privacy guarantee weakens as $b$ increases.*

**Amplification via shuffling:**  Fix a per-round batch size $B$. Then, suppose we shuffle the list of examples, and cyclically iterate over batches of size $B$ in this list as the sets of examples to use in each round of matrix factorization. That is, we shuffle $D$ into an ordered list $d_1, d_2, \ldots$, and in round $i$ use examples $d_{(i-1)B+1 \pmod m}, d_{(i-1)B+2 \pmod m}, \ldots, d_{iB \pmod m}$.

For simplicity let's consider the case where $m/(Bb)$ is integer. In particular, this means in this shuffling scheme, each example appears once every $m/B$ rounds, and for each of these rounds $i$, $i \pmod b$ is the same. Then this shuffling scheme is equivalent to the following: First, rather than choose an arbitrary partition to apply Thm. 4, we choose a uniformly random partition into $b$ subsets of size $m/b$. Then, we choose $\mathcal{S}$ to be the distribution giving by shuffling $[m/b]$ and then cyclically iterating over the shuffled list in batches of size $B$. Given this equivalence, we get the following:

**Corollary E.2.** *Suppose the examples in matrix factorization are chosen by shuffling $D$ and then iterating over batches of size $B$. If $n/(Bb)$ is integer, then the matrix factorization mechanism satisfies any standard privacy guarantee satisfied by $k$ adaptive scalar queries with sensitivity $\max_{i \in [n]} \|\mathbf{C}\mathbf{e}_i\|_2$ and noise $N(0, \sigma^2)$, with the examples in each query given by shuffling a dataset of size $m/b$ and cyclically iterating over this list in batches of size $B$.*

**Example E.2.** *Consider the simplified case where $m = n$, we choose a random permutation $\pi$, and in round $i$ query example $d_{\pi(i)}$. In this case, if all the column norms of $\mathbf{C}$ are 1, $\mathbf{x}$'s rows have sensitivity 1, and $\mathbf{z}$'s entries have standard deviation $\sigma = \mathcal{O}\left(\frac{\sqrt{\ln(1/\delta)}}{\epsilon}\right)$, we get that $\mathbf{C}\mathbf{x} + \mathbf{z}$ satisfies $(\epsilon, \delta)$-DP. With e.g., the amplification for shuffled $(\epsilon, \delta)$-DP mechanisms given by Theorem 5.1 of [6] and Cor. E.2, if $\epsilon$ is a constant, we instead get that $\mathbf{C}\mathbf{x} + \mathbf{z}$ satisfies $\left(\epsilon \cdot \mathcal{O}\left(\sqrt{\frac{b\log(1/\delta)}{n}}\right), \delta \cdot \mathcal{O}\left(\frac{n\ln(1/\delta)}{b}\right)\right)$-DP.*

## E.4 Additional Figures

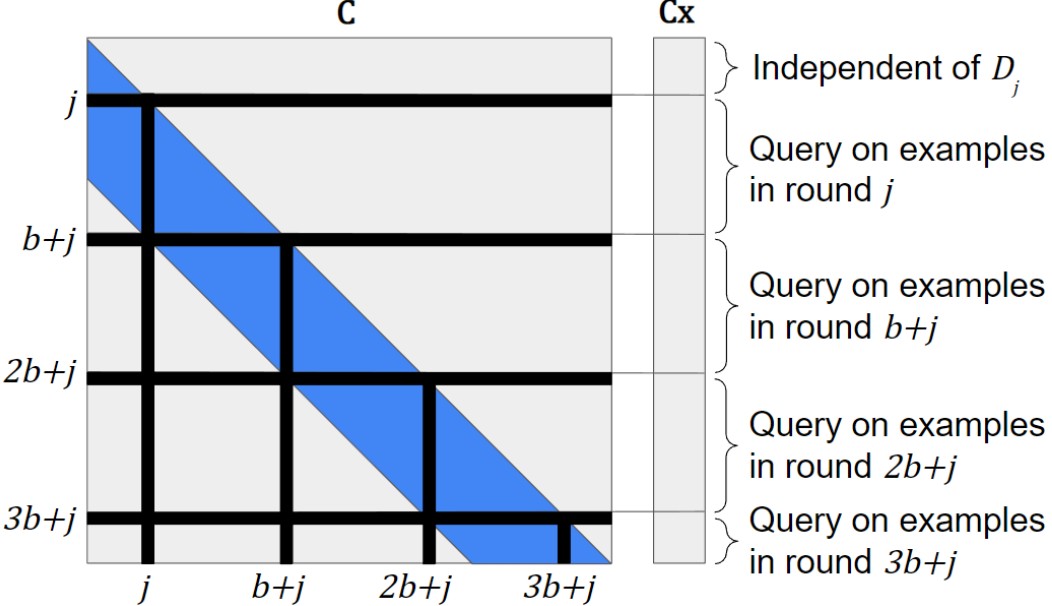

Figure 8: A visualization of how we can decompose a banded matrix mechanism into independent queries on $D_j$ (as in Algorithm 6) under our sampling scheme.

## F Additional RMSE Experiment Details