# OpenReview forum: "(Amplified) Banded Matrix Factorization: A unified approach to private training"
_NeurIPS.cc/2023/Conference — NeurIPS 2023 poster_

### Official Review · Reviewer_UhEM · 2023-06-20

**Soundness:** 3 good
**Presentation:** 3 good
**Contribution:** 2 fair
**Rating:** 6
**Confidence:** 3

**Summary:**

Applying matrix factorization mechanism and balancing tradeoffs of the mechanism in differential privacy is a long living issue. This work constructs MF mechanism with banded matrices for both centralized and federated training setting across all privacy budgets. For federated setting, this work is compatible with real world federated learning devices. For centralized setting, banded matrices are on par with the same privacy amplification results of DP-SGD algorithm.

**Strengths:**

1. Banded matrix factorization with amplification can outperform DP-SGD with amplification for centralized setting, and well-balance privacy-utility tradeoffs.
2. With b-min-sep-participation, banded MF can optimize (k,b)-participation, and reduce linear memory/time complexity for per-iteration noise generation to a constant.

**Weaknesses:**

1. The bound of sensitivity of MF mechanism looks confusing although the first part looks tight (sup for C*x and C*x_tilde). See Question 5.
2. Although this work considers FL in the practice, the dataset like CIFAR 10 seems quite outdated and the scale of CIFAR10 is too small, which is not enough to convince the community.
3. Probably authors need to clarify questions I stated in the next sections.

**Questions:**

1. Can you compare advantages and disadvantages of banded and unbanded matrix factorization for DP in FL setting?
2. In the Theorem 1, what is n? Is n the number of iterations or other things else?
3. In the Figure 1, it is hard to tell the difference of performance of Banded MF w/ amplification and multi-epoch MF w/o amplification. Why do you need amplification? What does the benefit of amplification for banded MF?
4. What does the \zeta mean for multiple participations? How does the \zeta affect the participations?
5. What is relationship between x, x tilde and u in equation 2?

**Limitations:**

1. This work has limited contribution on DP, because this work is more like an incremental work for multiple-epoch MF-DP-FTRL based on Choquette-Choo et al’s work to extend to federated learning setting.
2. It is not clear why we should assume in the cross-device FL setting, (k,b)-participation is not feasible. I know it is good to have some assumptions in the real world. However, I am wondering if it is necessary to restrict the scenario into such the practical case.
3. The MF optimization problem looks interesting and promising, but this work still needs more experiments, especially more datasets (at least three and other than small datasets like CIFAR10).

---

> ### Author Rebuttal · Authors · 2023-08-09
>
> Thank you for your detailed review.
>
> >**(1) Advantages and disadvantages of BandMF in FL:**
>
> Most importantly, the only prior unbanded approach to be applicable to cross-device FL is the single-epoch work, which we significantly outperform Empirically, we show this on  SO-NWP, a large-scale, non-iid, and common FL benchmark in Figure 5 (a) as well as experiments in production setting in Figure 5 (b).
>
> >**(2) What is n in Theorem 1:**
>
> It represents the number of iterations/steps of SGD. We will add this to the statement.
>
>
> >**(3) Benefits of Amplification in Figure 1:**
>
> Without amplification, Multi-Epoch Matrix factorization performs worse than DP-SGD with amplification for small $\epsilon$. This was shown in their work [14] and can be observed from Figure 1 (a) by comparing DP-SGD and Multi-Epoch MF. This is because in the small $\epsilon$ regime, the benefits of amplification are larger than the benefits of correlated noise, and prior to our work there was no proven privacy amplification guarantee for DP-MF. Proving such a guarantee is thus what  enables us to significantly outperform DP-SGD in a larger regime of $\epsilon$, and perform no worse than DP-SGD for all $\epsilon$. Amplification allows us to obtain tighter privacy budgets $\epsilon$ given the same protocol by accounting for how randomness in sampling can obscure data more. Importantly, this means that before our work, practitioners would need to implement both algorithms and check empirically which worked better for their setting; our work provides a single algorithm that is never worse, and often better.
>
> We will add the following explanation to the introduction after we reference Figure 1:
> “
> Empirically, we show that MF with amplification has privacy-utility tradeoffs that are no worse than DP-SGD for all, and often significantly betteras can be seen in Fig 1. This is in contrast to prior ME-MF work [14] which can be observed by DP-SGD outperforming it Fig 1.
> “
>
> >**(4) What does the \zeta mean for multiple participations? How does the \zeta affect the participations?:**
>
> Let's consider example-level DP / centralized training for simplicity. Then,  $\zeta$ is the clip norm which bounds the per-example gradient $\ell_2$ norm, exactly as in DP-SGD (see Alg 1). Thus, this controls the sensitivity only of that individual participation, and we require e.g. Thm 2 or Thm 3 to bound the total sensitivity. Note this is a different approach than DP-SGD, which does not explicitly bound the sensitivity to multiple participations, but instead uses DP composition across each participation (or amplification-via-sampling composition when sampling is done, and a fixed limit on participations is not present).
>
> Thus $\zeta$ does not influence the participation pattern, but does influence the final total sensitivity.
>
> >**What is the relationship between $x$, $\tilde{x}$, and $u$ in equation (2)?**:
>
> **x** represents the stream of gradients, i.e., the outputs of SGD throughout ML training, *that satisfy the participation pattern $N$*. For example, in multi-epoch MF of [14], this would be all sequences where there are at most *k* non-zero entries that are separated by exactly *b* steps. **$\tilde{x}$** represents the neighboring stream of gradients, i.e., replace all contributions by that single example with 0. This is required to bound the sensitivity as in DP analysis. This can be equivalently expressed as sampling **$u \in \mathfrak{D}$** if we define this **$\mathfrak{D}$** to be the set of all deltas **$x - \tilde{x}$**. This is described above equation (2), but are happy to make changes at your recommendation to improve the clarity.
>
>
> >**Limitations of Choquette-Choo et al. in the cross-device FL setting:**
>
> The work of Choquette-Choo et al. [14] is not applicable to the cross-device FL setting, which we discuss in the introduction and the beginning of Section 3. Thus, there is a significant gap in performance in this setting, as can be observed in Figure 6 (a). We get over 1.5 percentage points in improvement over the existing state-of-the-art (Single-epoch MF), which showcases that our work leads to significant practical improvements that could not be ascertained otherwise.
>
> >**Benefits of Amplification and Figure 1:**
>
> We have separated out our main contributions between those for cross-device FL and those for centralized settings. Please see the global response.

---

> > ### Comment · Reviewer_UhEM · 2023-08-13
> >
> > Thanks for your rebuttal.

---

### Official Review · Reviewer_4QAc · 2023-07-05

**Soundness:** 2 fair
**Presentation:** 3 good
**Contribution:** 2 fair
**Rating:** 5
**Confidence:** 2

**Summary:**

In this paper, the authors show how Matrix Factorization (MF) can subsume prior
state-of-the-art algorithms in both federated and centralized training settings, across all privacy
budgets. They apply the key technique: MF mechanisms with banded matrices. For both the
cross-device federated learning setting and the centralized setting, the result in this paper
improves or generalizes previous results. In addition, the $\hat{b}$-banded matrices
substantially improve the memory and time complexity.

**Strengths:**

1. The problem of Matrix Factorization for Differential privacy is well-motivated.
2. This paper is solid, the results look correct.
3. The result improves or generalizes previous results.

**Weaknesses:**

The main result for the centralized setting is summarized in Section 5. However, it seems the
result for federated learning setting is not stated clearly. Is there a summary of the result for FL
setting?

**Questions:**

See weakness.

---

> ### Author Rebuttal · Authors · 2023-08-09
>
> Thank you for your review and positive feedback.
>
> Our main results for the cross-device FL setting are written in sections 3 and 4. To make this clearer, we have rewritten our contributions section in the introduction to clearly state the contributions separately for cross-device FL and for centralized training. The new text is in the global response.
>
> We are happy to engage in more discussion if desired.

---

> ### Comment · Area_Chair_es4z · 2023-08-21
> **check the rebuttal**
>
> Dear Reviewer 4QAc,
>
> Could you please check the authors' rebuttal and respond to them? Also, please ask the authors if you have any follow-up questions.
>
> Sincerely,
> AC

---

### Official Review · Reviewer_T4Mk · 2023-07-06

**Soundness:** 3 good
**Presentation:** 3 good
**Contribution:** 3 good
**Rating:** 5
**Confidence:** 2

**Summary:**

This paper proposes banded matrix factorization for differential privacy. It shows this new mechanism can be effectively applied to centralized and federated-learning settings (where individuals can choose when and how many times to participate in training). The main technical part is the $b$-minsep-participation schema that generalizes the $(k, b)$-participation proposed in Choquette-Choo et al.


**Strengths:**

The experiments in Sec 6 demonstrate that banded MF exhibits a very good improvement over multi-epoch MF and DP-SGD in the centralized settings, and almost coincides with (or improves modestly) the state-of-the-art performance in the federated learning.

**Weaknesses:**

While the presentation of the paper is good for the most part, there are some issues that can be improved. For instance, the notation $\hat b$-banded matrices were used several times in the Abstract, Introduction, and Sec 2 without a precise (or even intuitive) definition [It was defined on Page 5!]. Also, $\hat b$ and $b$ seem to be interchanged arbitrarily (which is really confusing); see for instance lines 62, 188, 189. (I understand that $\hat b$ is taken to be equal to $b$ only in Sec 5, but not necessarily in the first 4 sections)
Also, Theorem 1 is rather cryptic: Do the "equal-sized subsets" mean batches of the same size? Does the partition change for each iteration (like typical SGD)? Is it not the case that $B b = m$?

This sentence (line 89-90) "The connection between DP empirical risk minimization [3, 4, 5, 7, 8, 9, 13, 15, 21, 23, 30, 34, 36,90 40, 45, 46, 47, 49] and DP online regret minimization [2, 4, 5, 24, 31, 33] has been studied for a long time." is awkward. What is the point of this sentence? Why not directly citing some of those works that studied this connection!!

What is SGDM in lines 95 and 101?

The argument in line 213 is not clear: How does Theorem 2 imply that $\text{diag}(X) = 1$ and bandedness mean the squared sensitivity is equal to $k$?



**Questions:**

Can the authors clarify the argument given in line 224: "Since we only have equality constraints on 224 individual entries of $X$, Problem 2 is essentially an unconstrained optimization problem"?
 Are you saying that any constrained optimization problems with equality constraints are essentially unconstrained?

**Limitations:**

Yes

---

> ### Author Rebuttal · Authors · 2023-08-09
>
> Thank you for your detailed review, and for providing line numbers in your review. We greatly appreciate it!
>
> >**$\hat{b}$ and b:**
>
> Thank you for highlighting this issue.  We now informally define $\hat{b}$-banded matrices in the abstract, and clarify the difference between b-min-sep and $\hat{b}$-banded matrices around Theorem 1 in the main-text, including forward pointers to their visualizations. We have also checked that our use of b vs $\hat{b}$ is consistent, using b to always refer to separation, and $\hat{b}$ to refer to a number of bands not necessarily equal to separation; this is also now noted in the notation table. The reason the two can be taken to be the same in Section 5 is that setting $b=\hat{b}$ is optimal for reasons that culminate in section 5.
>
> >**Theorem 1 “equal-sized subsets”:**
>
> The subsets are separate from the notion of batches in ML; we first partition the dataset into non-overlapping subsets, and then sample batches from within each subset to perform each round of (DP-)SGD. The partitioning into these subsets is fixed across all iterations. We will add the following example to the text, which we believe will clarify both of these points:
>
> “As an example, consider doing 2000 iterations of DP-SGD on CIFAR-10 (which has 50,000 examples) using a minibatch of 500 examples in each round of DP-SGD. This instantiation of DP-SGD has the same DP guarantees as answering 2000 queries using a subsampled Gaussian mechanism with sampling probability 500/50000 = .01.
>
> If we instead use, e.g., 10-banded DP-MF, our suggested sampling scheme is the following: Split CIFAR-10 into 10 subsets of 5000 examples each, D_1, D_2, … D_10. In rounds 1, 11, 21… we sample 500 examples from D_1, in rounds 2, 12, 22… we sample 500 examples from D_2, and so on. We sample from each subset a total of 2000/10 = 200 times, and each time our sampling probability is 500/5000 = .1 within the subset. So Theorem 1 shows this DP-MF satisfies the same DP guarantees as answering 200 queries with a subsampled Gaussian mechanism with sampling probability .1.“
>
> >**What is SGDM in lines 95 and 101?**
>
> Momentum-SGD, which we have clarified in the main-text.
>
> >**How does Theorem 2 imply diag(X) = 1 (Line 213)?**
>
> Theorem 2 guarantees that the sensitivity will be bounded by k\sqrt{k’}. Recall that k is an upper bound on the column norms of C. Then diag(X) = 1 guarantees that k=1 and so by theorem 2, the squared sensitivity is bounded by k’^2. We did have a typo in the main-text which will be corrected: (k’ instead of k).
>
> >**Unconstrained Optimization with Equality Constraints (Line 224):**
>
> Yes, because the equality constraints exactly specify some entries in X, we can simply consider a lower-dimensional unconstrained optimization that only considers the remaining entries in X. We have improved the wording in the paper to make this clear.
>
> >**Benefits in cross-device FL setting:**
>
>  Thank you for highlighting our strong empirical performance in the central settings. We would like to clarify that we do also outperform significantly in the cross-device FL setting (and not just coincide) because we actually report a **tighter privacy budget for our work**. Thus, we achieve (small) benefits in utility as you noted, but significant benefits in privacy. For reference, we reduced the privacy budget from 6.69 to 4.35.
>
> >**Citations in Related Works:**
>
> The point of these two lines (along with the citations) was to emphasize that there is a long line of work trying to understand the relationship between DP-online learning and DP-empirical risk minimization (DP-ERM). However, whether there is a tight connection between them is still an open question. Only recently, Asi et al. showed that DP-FTRL (a specific instantiation of the algorithmic framework in this paper) achieves the best known online learning regret. So, a natural question is whether our algorithmic framework is strong enough to obtain optimal DP-ERM/ excess population risk bounds purely from an online learning algorithm.
>
> We will clarify this point more in the paper.

---

> > ### Comment · Reviewer_T4Mk · 2023-08-21
> > **Thanks for the rebuttal**
> >
> > I thank the authors for the detailed response.
> >
> > All my concerns/questions have been addressed.

---

### Official Review · Reviewer_z4EU · 2023-07-06

**Soundness:** 4 excellent
**Presentation:** 4 excellent
**Contribution:** 4 excellent
**Rating:** 8
**Confidence:** 3

**Summary:**

The authors present a novel mechanism explicitly designed for differentially private training. The mechanism considers the sensitivity of different participation schemes in the context of fixed datasets during differentially private training. The key contributions of this work can be summarised as follows:
1. By accounting for the sensitivity of multistep participation schemes, the proposed mechanism eliminates the need for composition.
2. The authors illustrate how matrix factorisation can benefit from privacy amplification through subsampling and shuffling.
3. They propose a computationally efficient and precise implementation utilising b-banded matrices to compute sensitivity.
4. Additionally, they introduce an optimisation technique to craft these b-banded matrices.

The authors provide theoretical proofs and compelling experimental results to support their claim that their approach outperforms DP-SGD accounting.


**Strengths:**

* The paper effectively presents the problem, the proposed solution has a nice and clear structure, and the arguments flow in a natural way.
* The related work is well-presented, providing context and the existing related work gap, namely if the matrix factorization can benefit from the same privacy amplification techniques as DP-SGD.
* The experimental section is detailed, offering thorough explanations and interpretations of the results.
* The paper elegantly recovers privacy amplification bounds from DP-SGD by carefully constructing the proposed linear operator.

**Weaknesses:**

* the current presentation is not self-contained, as it has plenty of references to the appendix/other related work. It is not ideal, as the main statements of the paper are a bit hard to fully understand without them. I know that space is a limiting issue, especially for well-detailed works like this, but please reconsider restructuring so that the main results (Theorem 2. and Theorem 4.) are possibly more easily to grasp without relying on the appendix this much.


**Questions:**

* More of a high-level question for the authors. In this scenario in which composition is bypassed altogether, why is the Gaussian Mechanism the "defacto" mechanism? I might miss something, but I always associate the Gaussian mechanism with composition + l2 sensitivity. It seems the authors have some space to explore other mechanisms (possibly). I was curious if the authors thought more in detail about this; is the Gaussian mechanism optimal in this scenario?
* A unique perspective on this paper is that most amplification techniques are derived by analysing the linear operator itself and not analysing the output distributions (common, for example, for the subsampled Gaussian mechanism). This is a very interesting view of this work, as it abstracts the noise distribution from the privacy amplification guarantees, giving a more general/decoupled view of privacy amplification in this case. Do the authors think it is possible to design linear operators further to improve privacy guarantees/utility tradeoffs (like other forms of amplification given by the operator's structure), or have we reached some optimal design?


**Limitations:**

* Increased  $O(b)$ multiplicative time/memory overhead compared to DP-SGD.

---

> ### Author Rebuttal · Authors · 2023-08-09
>
> Thank you for your positive feedback, comments, and questions.
>
> >**Make Paper Self-Contained:**
>
> Thank you for these suggestions. We used the additional page to add in an informal proof to Theorem 2 and to add algorithm 4 to the main-text, with a description of how it works. We also have expanded on details elsewhere that may have been pushed to the appendix. Please see the global response describing how we restructured Theorem 4 / Section 5 so that section is self-contained.
>
> >**Optimizing Privacy-Utility Tradeoffs:**
>
> We agree that the problem of choosing C with the benefits of amplification in mind is interesting and merits future work. Note that, if only using the amplification statement in this paper, the approach in “Optimizing the number of bands” is optimal under the diag(X) = 1 constraint: the resulting privacy guarantee only depends on n (fixed), the number of bands (we can try all choices simultaneously), and the column norms of C (fixed by the diag(X) = 1 constraint). Furthermore, under our sampling scheme this amplification statement is tight, since it captures a tight analysis for DP-SGD. It may be possible for a different sampling scheme one can derive an amplification statement that depends more on the structure of C, but deriving this amplification statement and optimizing C given this statement are likely to be difficult.
>
> >**Gaussian Mechanism Optimality:**
>
> The fact that Gaussian mechanism is the optimal noise addition algorithm in our setting (with a generic choice of C, and one privatizes Cx as Cx + noise) stems from the fact that the $\ell_2$-sensitivity of the complete mechanism (w.r.t. a single user) is bounded. (Results like [BUV14] have shown matching lower bounds.) Regarding the comment about avoiding composition, it is well known that (modulo adaptivity of the algorithm) the bound attained by bounding the overall $\ell_2$ sensitivity, can also be achieved by [zCDP] composition and vice versa. Hence, the views are isomorphic. In addition, it is easier to work with Gaussians in the adaptive setting as prior works have shown that in Gaussian noise addition mechanism for SGD style algorithms, there is no difference between an adaptive and non-adaptive algorithm; this is not true for all mechanisms, as demonstrated by Appendix D.1 of [Denisov et al. 2022] .
>
> [BUV14] Bun, Mark, Jonathan Ullman, and Salil Vadhan. "Fingerprinting codes and the price of approximate differential privacy." Proceedings of the forty-sixth annual ACM symposium on Theory of computing. 2014.
>
> [zCDP] Bun, Mark, and Thomas Steinke. "Concentrated differential privacy: Simplifications, extensions, and lower bounds." Theory of Cryptography Conference. Berlin, Heidelberg: Springer Berlin Heidelberg, 2016.
>
> [Denisov et al. 2022] Denisov, Sergey, et al. "Improved differential privacy for sgd via optimal private linear operators on adaptive streams." Advances in Neural Information Processing Systems 35 (2022): 5910-5924.

---

> > ### Comment · Reviewer_z4EU · 2023-08-14
> >
> > Thank you for your responses, and looking forward to the internal discussions with other reviewers!

---

> > ### Comment · Reviewer_4QAc · 2023-08-21
> >
> > Thanks a lot for the response. I will keep my score.

---

### Official Review · Reviewer_2XT3 · 2023-07-23

**Soundness:** 3 good
**Presentation:** 1 poor
**Contribution:** 2 fair
**Rating:** 6
**Confidence:** 3

**Summary:**

The paper studies the problem of how to optimize the Matrix factorization (MF) mechanisms so that the effect of random noise can be minimized. The MF mechanisms can be applied in addition to the well-known DP-FTRL or similar online DP algorithms in machine learning training. This technique decomposes the query matrix A (e.g., a matrix encoding the prefix sum) into two matrices and adds noise to the intermediate matrix multiplicative result of the data and one of the decomposed matrices. It can potentially introduce less variance to the final results. The key idea of the paper is formulating the MF problem into an optimization problem. The authors show the sensitivity of the mechanism under some constraints of the participation schema and discuss how to derive the best decomposition in terms of minimizing the error profile because of the decomposition. Some experimental results are provided to demonstrate that the proposed mechanism is at least as good as the classic DP-SGD mechanism.

**Strengths:**

Generally speaking, this paper provides some in-depth results on the MF mechanism that can be interesting to the community.
1. The authors provide an insightful analysis angle on how to optimize the MF mechanism.
2. The authors derive the sensitivity of the MF mechanism under some additional assumptions about the participation schema of the users and the property of the original query matrix.
3. The authors show how one should optimize the MF mechanism via decomposition and how the mechanism can enjoy privacy amplification via sub-sampling.


**Weaknesses:**

1. While the technical contributions are sufficient for building a good paper, the presentation in this paper makes readers somehow hard to follow.

- MF mechanism is relatively new compared with the other DP mechanisms. So MF may be ambiguous if a reader is not familiar with the ancestors of this work (i.e., [14, 16]), as some readers may be confused DP MF problems (e.g. [1*, 2*]) with the MF mechanism at the first glance. Introducing problem formulation in (the beginning of) the introduction section may help readers evaluate and understand the value of the paper.
- The explanation of why the MF mechanism can help and how the proposed methods are derived in this paper is not easy to follow.
- It is unclear what are benefits one can get by following the participation schema and decomposition method proposed methods. There may be some discussion scattered around the paper, but a summary (via a table) comparing the results of this paper to the existing ones may deliver the results more directly to the readers.
- The key algorithms are most deferred to the appendix. So main text itself is not self-contained and reader-friendly (readers have to jump between the main text and the appendix). At least some algorithms closely related to the main contribution of the paper, like the sampling scheme, should be stated in the main text, not just the theoretical results using this algorithm.

2. Some experimental results may need further explanation.
- Why the BANDMF and  MULTI-EPOCH MF tends to introduce a similar level of noise as DP-SGD when the number of epoch increases?
- What is the exact RMSE for BANDMF, MULTI-EPOCH MF and DP-SGD in Figure 4? Also, how should one understand the relationship between RMSE and accuracy (From Table 5 in appendix, smaller noise in DP-SGD can have worse performance than ampl MF)?


[1*]Hyejin Shin, Sungwook Kim, Junbum Shin, and Xiaokui Xiao. 2018. Privacy-enhanced matrix factorization for recommendation with local diferential privacy. IEEE Transactions on Knowledge and Data Engineering 30, 9 (2018), 1770–1782.

[2*] Zitao Li, Bolin Ding, Ce Zhang, Ninghui Li, and Jingren Zhou. Federated Matrix Factorization with Privacy Guarantee. PVLDB, 15(4): 900 - 913, 2022

**Questions:**

Refer to the Weaknesses

---

> ### Author Rebuttal · Authors · 2023-08-09
>
> Thank you for your comments, suggestions and questions.
>
> >**MF may be ambiguous**
>
> This is an excellent point that we missed. We have added the following footnote on the first page: “
>
> “We use matrix factorization for the algorithmic design of DP mechanisms, and then use these mechanisms in general-purpose gradient-based private optimization algorithms. This use of matrix factorization is unrelated to machine learning settings such as collaborative filtering or recommender systems where the ML problem itself involves matrix factorization [37, 41, 51].
> “
>
> [37] Koren, Yehuda, Robert Bell, and Chris Volinsky. "Matrix factorization techniques for recommender systems." Computer 42.8 (2009): 30-37.
>
> [41] Zitao Li, Bolin Ding, Ce Zhang, Ninghui Li, and Jingren Zhou. Federated Matrix Factorization with Privacy Guarantee. PVLDB, 15(4): 900 - 913, 2022
>
> [51] Hyejin Shin, Sungwook Kim, Junbum Shin, and Xiaokui Xiao. 2018. Privacy-enhanced matrix factorization for recommendation with local diferential privacy. IEEE Transactions on Knowledge and Data Engineering 30, 9 (2018), 1770–1782.
>
> >**Why the MF mechanism can help**
>
> See the global response including the new figure, which is intended to address this question.
>
> >**The key algorithms are mostly deferred to the appendix:**
>
> Please see the global response; MF-DP-FTRL pseudocode is now given early in the paper, and we have also restructured Section 5 so it is self-contained. As we attempt to add clarity to the other areas where the presentation was unclear, it would be extremely helpful to have specific line numbers or specific questions for any remaining issues
>
> >**Also, how should one understand the relationship between RMSE and accuracy (From Table 5 in appendix, smaller noise in DP-SGD can have worse performance than ampl MF)?**
>
> Generally, lower RMSE in the estimates of cumulative sums of gradients correlates very well with learning accuracy (e.g., as demonstrated by its ability to predict good learning rates, Fig 11).  However, the noise multiplier sigma is not directly comparable between DP-SGD and MF: In DP-SGD this noise is added directly to the gradients, but in MF linear combinations of noise are added via $C^{-1}$, generally in such a way that some previous noise cancels out when gradients are summed across iterations (see also the new figure that illustrates this intuition). Thus, sigma is included in e.g. Table 4 and 5 for reproducibility (these are the actual parameters used to run the experiments), but not to directly facilitate algorithm comparisons.  We will clarify this in the paper.
>
> >**Importance of the Participation Pattern:**
>
> The participation pattern we introduce, $b-min-sep$-participation, enables the main results and claims of our paper for the cross-device FL setting. Without this participation pattern, we would not be able to bound sensitivity in this setting. Note that, for the centralized ML setting, this participation pattern is not needed because (k,b)-participation can be enforced and will also satisfy our amplification guarantees. We have expanded the discussion before Definition 1 to the following:
>
>
> “In Defn. 1, we propose b-min-sep-participation, a generalization of (k,b)-participation which can be practically enforced by cross-device FL systems, thus enabling us to leverage BANDED-MF in this setting (Section 5).\footnote{Note that, b-min-sep-participation is required for the cross-device FL setting. But, (k,b)-participation can be used in centralized settings where it can be enforced and it will also satisfy our BAND-MF amplification guarantees leading to the empirical results we show in Figure 1.} In b-min-sep-participation, the distance between any two participations is at least b, rather than exactly b as in (k,b)-participation:“
>
> >**Further explanation of experimental results**
>
> As the number of epochs changes, so does the sensitivity of the encoder matrix C, and subsequently the geometry of the optimization problem.  By equation 5, we see that sensitivity can be calculated as a sum of absolute values of entries of X corresponding to iterations where a single user might participate.  As the number of epochs increases, the number of entries of X that we have to sum up also increases.  It turns out that under the constraint of constant sensitivity, it is better to put more weight on the diagonal of X than the off-diagonal.  This is an observation we have made by solving this optimization problem numerically in a number of settings. Hence, as the number of epochs increases, X becomes more and more diagonally dominant, making the mechanism closer to DP-SGD (X = Identity).  We  have added a version of this discussion to Section G.
>
> The exact RMSE is shown in Fig 5(c), although we primarily looked at the relative RMSE between the different mechanisms in our interpretation of the results.  The exact RMSE is something we purposely omit in Fig 5(a-b) in favor of the RMSE ratio as it is a quantity that is more difficult to interpret directly.  One should generally expect RMSE and learning performance to correlate pretty well, as shown by [Kairouz et al, 2021].  Note that in Table 5, “sigma” should not be compared between mechanisms, as it has different interpretations for different mechanisms — the values are only included for reproducibility.  We added a comment on the header of table 5 to address this confusion, and we provide a more in depth discussion of this issue below.

---

> > ### Comment · Reviewer_2XT3 · 2023-08-14
> >
> > Thanks for the explanations and improvements on the paper. I am satisfied with the response, so I will raise my score.

---

### Author Rebuttal · Authors · 2023-08-09

Thank you for your time in reviewing our work.

>**On intuition and improved organization.**

We have revised the paper to briefly describe the matrix mechanism in the first paragraph, and have additionally included a new figure to help provide intuition for the advantage of MF-DP-FTRL, as well as complete pseudocode for both MF-DP-FTRL and DP-SGD early in the paper (see attached pdf). We have also added the following text to provide additional intuition for our view of MF in ML:

The (new figure) compares DP-SGD and MF-DP-FTRL. To gain an intuition for why MF-DP-FTRL can perform better than DP-SGD, observe that vanilla SGD has iterates $\theta_t = \theta_0 - \eta \sum_{i=1}^t \hat{x_i}$, and hence when the noisy gradients $\hat{x_i}$ are added, the $C_{i,j}^{-1}$ terms in MF-DP-FTRL serve to \emph{cancel out} some of the noise introduced on previous rounds. The noise cancellation reduces the total error in all the prefix sums of gradients $\sum_{i=1}^t \hat{x_i}$ for $t \in [n]$, but also worsens the privacy guarantee of the mechanism, i.e. increases its sensitivity. The privacy worsens as e.g., an adversary trying to learn $x_1$ via $\hat{x_1}$ can partially learn the value of $\mathbf{z_1}$ from $\hat{x_2}$, whereas in DP-SGD $\hat{x_1}$ and $\hat{x_2}$ are uncorrelated. Hence there is a tradeoff between the total error and sensitivity (see next paragraph): DP-SGD sets $C_{i,j}^{-1}= 0$ below the main diagonal, effectively minimizing sensitivity (assuming a fixed normalization of the main diagonal), but with a large total error due to no noise cancellation. On the other hand, MF-DP-FTRL can arrive at a better compromise between mechanism sensitivity and the total error. This is formalized in the optimization problem of Sec 4.

Without a sampling assumption, the implied DP adversary knows which examples participated in batch $S_i$, and for DP-SGD with uncorrelated noise, knows they only need to “attack” $\hat{x_i}$. However, with MF-DP-FTRL, the information from $S_i$ can potentially be masked with a larger amount of initial noise in $\hat{x_i}$, which is then canceled out over subsequent rounds.  “Spreading out” the release of information about batch $S_i$ over a larger number of iterations in this way can intuitively provide better privacy, while still allowing for accurate partial sums of gradients (and hence SGD iterates). This is, in a loose sense, similar to the way SGD with sampling "hides" information about a particular example at randomly chosen iterations.


>**On reducing references to Appendix from Section 5:**

We agree with the reviewers that Section 5 will be easier to read without references to the appendix. We have rewritten the section so that it only refers to the sampling scheme in Algorithm 2 which is already in the main body, rather than the more generic sampling scheme in the appendix. In doing so we have cut or replaced all references to Algorithm 5, which is in the appendix. We have also rewritten Theorem 4 to explicitly say “satisfied by the composition of k subsampled Gaussian mechanisms with sampling probability bp.” instead of “satisfied by Algorithm 6 in App E.1”. This removes the need to refer to the appendix to understand the main algorithms/ideas of this section, with minimal changes to the overall structure/flow.

>**FL versus centralized ML contributions:**

We have now separated out our main contributions between those for cross-device FL and those for centralized ML in the introduction. This is shown below.


“
**Contributions for cross-device FL:**
Here, the (k,b)-participation schema of Choquette-Choo et al. [15] cannot be enforced.  We propose a strict generalization, b-min-sep-participation, and show how to efficiently and exactly bound the sensitivity for banded matrices in Thm. 2, allowing formal DP guarantees and the numerical optimization of optimal mechanisms (Sec. 4). Culminating from this, and because b-min-sep-participation can be practically enforced in FL infrastructure, we show that this leads to significant privacy-utility benefits in a production deployment (Fig. 6 of Sec. 6). Our work also generalizes the sensitivity calculations of Choquette-Choo et al. [15] to provide a general upper-bound on b-min-sep-participation sensitivity (Thm. 3), which allows the matrices of Choquette-Choo et al.[15]to be used in the FL setting, as well as removing the need to exactly bound b before training (see Sec. 6 and App. K).

**Contributions for centralized training:** The existing privacy amplification analysis of DP-SGD does not allow for the correlated noise that is applied in MF-DP-FTRL. Our paper introduces a novel partitioning of the SGD iterates into independent queries A crucial observation is that one can argue the DP for each data sample index independently, which in turn allows us to use different partitioning views. This allows us to prove in Thm. 4 of Sec. 5 that banded matrices enjoy the benefits of privacy amplification, and show that DP-SGD is a special case, giving us the best of both algorithms. This enables us to always pareto-dominate DP-SGD, unlike Choquette-Choo et al.[15] which only does so for large enough as observed in Fig. 1.  Further, this allows us to improve on both baselines, between 1−4%-points. Informally, our amplification result is: ...
“

---

### Decision · Program_Chairs · 2023-09-21

**Decision:**

Accept (poster)

**Comment:**

In this paper, the authors propose a new optimization based method to reduce the effect of added noise for matrix factorization, and the proposed one can be applied to both centralized and federated scenarios. Analysis and experimental results demonstrate that the proposed one is better than (or at least as good as ) DP-SGD.  To this end, I recommend accepting this submission.

However, I do have some concerns regarding the writing and the potential impact on applications. For writing, it should be further improved as most of reviewers complain about it. I also suggest the authors to add more discussions about the potential applications to make it practical.

Hope the authors find the discussions with reviewers useful and make this submission a better one.